# NIR-dye bridged human serum albumin reassemblies for effective photothermal therapy of tumor

Zhaoqing Shi [1,2,4], Miaomiao Luo[1,4], Qili Huang[2,4], Chendi Ding[1,3], Wenyan Wang[2], Yinglong Wu[3], Jingjing Luo[2], Chuchu Lin[2], Ting Chen[1], Xiaowei Zeng[2], Lin Mei [1] ✉, Yanli Zhao [3] ✉ & Hongzhong Chen [2,3] ✉

Human serum albumin (HSA) based drug delivery platforms that feature desirable biocompatibility and pharmacokinetic property are rapidly developed for tumor-targeted drug delivery. Even though various HSA-based platforms have been established, it is still of great significance to develop more efficient preparation technology to broaden the therapeutic applications of HSA-based nano-carriers. Here we report a bridging strategy that unfastens HSA to polypeptide chains and subsequently crosslinks these chains by a bridge-like molecule (BPY-Mal$_2$) to afford the HSA reassemblies formulation (BPY@HSA) with enhanced loading capacity, endowing the BPY@HSA with uniformed size, high photothermal efficacy, and favorable therapeutic features. Both in vitro and in vivo studies demonstrate that the BPY@HSA presents higher delivery efficacy and more prominent photothermal therapeutic performance than that of the conventionally prepared formulation. The feasibility in preparation, stability, high photothermal conversion efficacy, and biocompatibility of BPY@HSA may facilitate it as an efficient photothermal agents (PTAs) for tumor photothermal therapy (PTT). This work provides a facile strategy to enhance the loading capacity of HSA-based crosslinking platforms in order to improve delivery efficacy and therapeutic effect.

HSA, the most abundant protein found in blood plasma, has been extensively utilized as a highly effective carrier for drug delivery since the advent of Abraxane, the pioneering HSA-based drug delivery system[1-5]. In spite of the recent emergence of several biocompatible carrier materials, few can match the well-established HSA-based therapeutic platforms that are proven in both bench and bedside. So far, HSA-based platforms have demonstrated proficiency in loading hydrophobic molecules, enhancing biocompatibility, extending circulation times, and devising drug delivery systems that are tailored or responsive to specific targets[6-8]. As an excellent carrier, HSA provides multiple surface modification sites (such as the thiol group provided by cysteine-34 and -NH$_2$ provided by N-doner residues) and hydrophobic domain, thus various kinds of drugs can be loaded to HSA via covalent coupling or non-covalent binding[9-11]. For example, Lu et al. recently reported an SN38-based albumin-binding prodrug, which was prepared through the Michael addition reaction between the maleimide group on SN38 prodrug molecules and the thiol group at the cysteine-34 residue[12]. Xie and coworkers developed a conjugated

[1]Tianjin Key Laboratory of Biomedical Materials, Key Laboratory of Biomaterials and Nanotechnology for Cancer Immunotherapy, Institute of Biomedical Engineering, Chinese Academy of Medical Sciences & Peking Union Medical College, Tianjin 300192, P. R. China. [2]School of Pharmaceutical Sciences (Shenzhen), Sun Yat-sen University, Shenzhen 518107, China. [3]School of Chemistry, Chemical Engineering and Biotechnology, Nanyang Technological University, 21 Nanyang Link, Singapore 637371, Singapore. [4]These authors contributed equally: Zhaoqing Shi, Miaomiao Luo, Qili Huang.
✉e-mail: meilin@bme.pumc.edu.cn; zhaoyanli@ntu.edu.sg; chenhzh58@mail.sysu.edu.cn

polymer that mimics fatty acids and can effectively interact with HSA to form nanoparticles via hydrophobic interaction[13]. Other than loading drugs directly, assembling HSA molecules to form HSA-based nanoparticles is also an emerging technique to fabricate HSA-based platforms[10,11,14,15]. Such an approach involves the use of crosslinking agents, such as glutaraldehyde, to prepare drug delivery materials with nanoscale dimensions by chemically crosslinking different HSA molecules together[16]. This strategy brings benefits such as enhanced tumor uptake efficiency mediated by albumin-binding receptor and enhanced permeability and retention (EPR) effect[17]. More ingeniously, the HSA-based nanoparticles assembled by the responsive linkers can respond to the tumor microenvironment (TME) to realize enhanced tumor targeting and penetration[18,19]. Nevertheless, the loading capacities of these HSA-based platforms are still relatively lower than other types of materials because of the insufficient binding sites of HSA[20,21]. For instance, a natural HSA molecule could only offer one free thiol group for drug coupling, thus, the loading efficacy of thiol modification strategies cannot meet the anticipation. Besides, the accessibility to most of HSA-drug conjugates-based nanomaterials require tedious synthesis and preparation steps, therefore they are too complicated to scale up for now. Therefore, it is still a challenge to realize their deep clinical translation due to the lack of strategies to construct albumin nanoparticles with high drug-loading capacity and facile preparation features.

Photothermal therapy (PTT), an emerging method that utilizes photo energy to generate heat and induce damage through photothermal agents (PTAs), has shown encouraging therapeutic effects for treating tumors with controllable precision[22-28]. In addition, the photothermal effect from photothermal therapy not only kills cancer cells but also generates sound waves that can provide additional diagnostic information through the process of photoacoustic (PA) imaging[25]. The most popular materials of PTAs include inorganic materials (such as gold nanomaterials[28-30], carbon-based nanomaterials[31,32], and MXenes[33,34]), conjugated polymeric materials[35,36], and small molecular dye PTAs[37-39]. Although these inorganic and conjugated polymeric materials exhibited great therapeutic performances, their clinical applications are severely restricted due to their non-degradable nature and latent toxicity risks. In comparison, small molecular dye PTAs (such as cyanine, porphyrin, boron dipyrromethene (BODIPY), and diketopyrrolopyrrole) are more metabolizable[37-39]. Among these PTAs, the aza-BODIPY have been widely explored as an NIR phototherapeutic agent owing to highly tunable photophysical and physicochemical properties[40]. Despite these merits, most of the small molecular PTAs are poorly soluble in water, for this reason, they must be formulated with drug carriers before administration[41-45].

Capitalizing on these current bottlenecks, herein, we report a bridging strategy to fabricate a novel HSA-based PTA nano-platform, which was composed of biocompatible HSA as the carrier scaffold and a rationally designed PTA dye molecule as the crosslinking agent. As illustrated in Fig. 1, the HSA was first reduced into multiple polypeptide chains. By this means, there is a large increase in the quantity of thiol groups as modification sites for further decoration with maleimide groups. In the meantime, we construct a bi-maleimide functionalized BODIPY dye as PTA (denoted as BPY-Mal$_2$), which could act as a bridge to crosslink the disassembled HSA chains into BPY@HSA nanoparticles through Michael reaction driving the assembling process (Supplementary Note 1). Our bridging strategy greatly simplifies the preparation procedures for constructing albumin-based therapeutic nanoplatform, since the therapeutic agent loading and the crosslinking steps are merged. More excitingly, this innovative design provides an

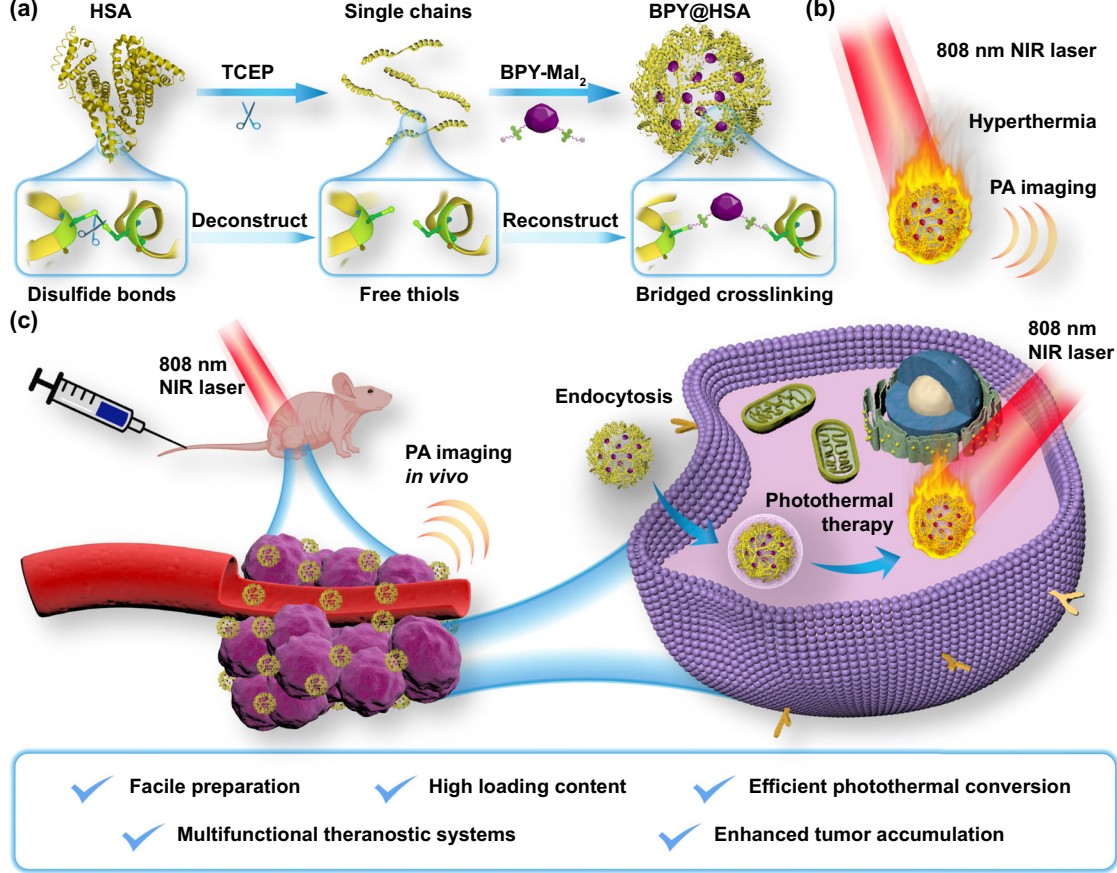

**Fig. 1 | Fabrication of BPY@HSA and its application in tumor treatment. a** Fabrication processes of BPY@HSA by the bridging strategy. **b** Therapeutic functions of BPY@HSA. **c** Schematic illustration of photoacoustic imaging and PTT on tumor model. Created with BioRender.com.

approach for creating albumin-based therapeutic nanoplatforms using a method of "deconstruction-reconstruction". The original disulfide bonds in albumin that contribute to the tertiary structure are cleaved first and subsequently replaced by PTA-based linkers that also hold therapeutic effect. In order to compare our bridging strategy with the conventional modification strategy, we also modified the pristine HSA with BPY-Mal$_2$ directly, which was denoted as BPY-HSA. It was found that our bridging strategy improved the loading capacity of HSA by orders of magnitude, and the as-prepared BPY@HSA presented better PTT activity and PA response than BPY-HSA. As a consequence, the BPY@HSA showed enhanced in vitro PTT treatment efficacy, and in vivo studies illustrated that BPY@HSA could effectively accumulate in tumor sites and serve as a safe PTA to photothermally ablate the tumors.

## Results

### Fabrication and characterization of BPY@HSA

The design philosophy of BPY@HSA was based on the bridge-like BPY-Mal$_2$ linker, which reconnected the thiol groups resulted from the cleaved disulfide bonds in HSA. In this study, we adopted BODIPY as the core of BPY-Mal$_2$, owing to its tunable optical properties in the NIR region, good chemical and photic stabilities, high extinction coefficient, and facile synthetic feature[40]. The BPY-Mal$_2$ was prepared according to the synthetic approach shown in Supplementary Fig. 1. Detailed synthesis and characterization information can be found in the Supplementary Information (Section S1) and Supplementary Figs. 2–11. The formulation containing only BPY-Mal$_2$ was prepared by the antisolvent precipitation method, affording an aqueous dispersion of BPY-Mal$_2$. This dispersion sample was denoted as BPY. The photothermal conversion capability of BPY was evaluated first to ensure it could work as PTA. As shown in Supplementary Fig. 12, BPY exhibited a significant temperature increase ($\Delta T = 60\,°C$) at $50\,\mu g/mL$ under 808 nm NIR laser irradiation ($1\,W/cm^2$). Encouraged by its photothermal ability, we further fabricated BPY-HSA and BPY@HSA via chemical binding or bridging strategy, respectively. The detailed fabrication difference between BPY-HSA and BPY@HSA was demonstrated in Supplementary Fig. 13.

The as-prepared BPY@HSA and BPY-HSA exhibited good water solubility (Supplementary Fig. 14). Transmission electron microscopy (TEM) images demonstrated the granular morphology with a size of around 20 nm of BPY@HSA (Fig. 2a, Supplementary Fig. 15), which was similar with the average hydrodynamic diameter ($38.2 \pm 1.7$ nm) in aqueous solutions measured by dynamic light scattering (DLS), as shown in Fig. 2b. The hydrodynamic diameter of BPY-HSA ($6.09 \pm 0.41$ nm) was similar to that of natural HSA (Supplementary Fig. 13). When BPY@HSA was incubated in water or in DMEM complete medium (containing 10% of FBS), the size of BPY@HSA showed negligible changes for up to 7 days (Supplementary Fig. 16), indicating the good stability of BPY@HSA, which is important for further in vivo application. The zeta potential values of BPY, BPY-HSA, and BPY@HSA were measured to be $28.0 \pm 2.1$ mV, $-38.4 \pm 0.5$ mV, and $-38.9 \pm 0.5$ mV, respectively (Fig. 2c). The negative shift in surface charges of BPY-HSA and BPY@HSA indicated that BPY-Mal$_2$ was encapsulated into the formulations. Furthermore, we determined the loading efficiency (LE) and loading capacity (LC) as shown in Fig. 2d, and the LEs of BPY-HSA and BPY@HSA were 77.7% and 29.6%, respectively. Notably, the LC of BPY@HSA (26.1%) was 9-times higher than that of BPY-HSA (3.0%), which was also higher than most of the reported HSA-based platforms by orders of magnitude (Supplementary Table 1). As compared with those formulation strategies, BPY-Mal$_2$ in our strategy not only serves as the crosslinking agents, but also acts as the therapeutic agent of the systems, since the therapeutic agent loading and the crosslinking steps are merged. Therefore, our methodology offers new avenues for formulating HSA-based therapeutic platforms, which are both feasible and straightforward.

As shown in Fig. 2e, BPY-HSA and BPY@HSA exhibited similar characteristic absorption peaks compared with BPY, suggesting that the BPY-Mal$_2$ could be successfully loaded by both methods. With the confirmation that the incorporated BPY-Mal$_2$ could convert the incident NIR light into heat efficiently, we further tested the photoacoustic responses of BPY, BPY-HSA and BPY@HSA. All of them exhibited bright photoacoustic signals (Fig. 2f) in dose-dependent responses ($R^2 > 0.9$, Supplementary Fig. 17). The photothermal conversion performances of BPY-HSA and BPY@HSA were subsequently investigated by a thermal imaging system (Fig. 2g–l, Supplementary Fig. 18). Upon irradiation by 808 nm NIR light, the BPY-HSA and BPY@HSA exhibited concentration- and power-dependent photothermal temperature elevation behavior. They also presented a high photothermal stability during five heating-cooling cycles, and there are negligible changes in the photothermal performance. The photothermal conversion efficiency (PCE) of BPY-HSA and BPY@HSA were calculated by fitting the heating-cooling curves, which were 61% and 56%, respectively, indicating that they had potent as PTAs for efficient PTT.

### In vitro cellular uptake and infiltration in 3D tumor spheroids

Owing to overexpressed albumin-binding proteins in tumor cells and the nutrient delivery function of HSA, HSA-based therapeutic platforms have a unique advantage for tumor uptake[46]. To investigate the cellular internalization performances of BPY@HSA, we labeled it by maleimide functionalized Cy5 (Mal-Cy5), and denoted it as Cy5-BPY@HSA. HSA was also marked by Mal-Cy5 as a reference group (Cy5-HSA) to replace BPY-HSA since there was no extra free thiol group in BPY-HSA for Mal-Cy5 modification. The in vitro cellular uptake of Cy5-HSA and Cy5-BPY@HSA in MCF-7 and 4T1 cells at different time were evaluated by flow cytometry (FCM) and confocal laser scanning microscope (CLSM). As shown in Fig. 3a–d, the fluorescence signals of Cy5 became stronger by prolong the incubation time, suggesting that both Cy5-HSA and Cy5-BPY@HSA could be internalized by MCF-7 and 4T1 cells. According to further quantification analysis results (Supplementary Fig. 19), the mean fluorescence intensity (MFI) in Cy5-BPY@HSA groups were higher than the Cy5-HSA groups, which indicated that BPY@HSA exhibited better cellular uptake efficiency. The CLSM images indicated the fluorescence signals in the cytoplasm were more obvious in Cy5-BPY@HSA treated groups than the Cy5-HSA treated groups (Fig. 3e, f), which was in accordance with the FCM results. Cellular photoacoustic imaging results also verified that BPY@HSA possessed better tumor cell uptake capability than BPY-HSA (Fig. 3g). MCF-7 and 4T1 cells treated by BPY@HSA showed much brighter photoacoustic signals than the BPY-HSA treated groups.

In order to explore the reason why Cy5-BPY@HSA exhibited better cellular uptake efficiency, we investigated the uptake mechanism of natural HSA and our bridged nanoparticles. As shown in Fig. 3h, i and Supplementary Fig. 20, the uptake of both formulations was greatly involved in energy-dependent pathways as 4 °C treated cells presented the lowest uptake, indicating active transportation processes on the uptake of Cy5-HSA and Cy5-BPY@HSA formulations. In particular, obviously decreased uptake efficacy on mβ-CD groups suggested that gp60 was the essential receptor affecting the uptake of the two formulations (Cy5-BPY@HSA formulation in especial), and the caveolae and clathrin-independent carriers (CLIC) also act as potential pathways of the two formulations in cellular uptake (evidenced by Simvastatin, Supplementary Fig. 21). The dynamin-dependent endocytosis also served as an efficient pathway in cell uptake, as clathrin-independent/dynamin-dependent endocytosis (FEME) was evidenced by the decrease in Genistein and Dynasore treated groups, and clathrin-mediated endocytosis (CME) was evidenced by the decrease in Dynasore and Chloroquine treated groups. Amiloride-treated groups demonstrated that both of the formulations were involved in macropinocytosis, and Cy5-BPY@HSA nanoparticle formulations relied more on macropinocytosis. Interestingly, the uptake efficacy on

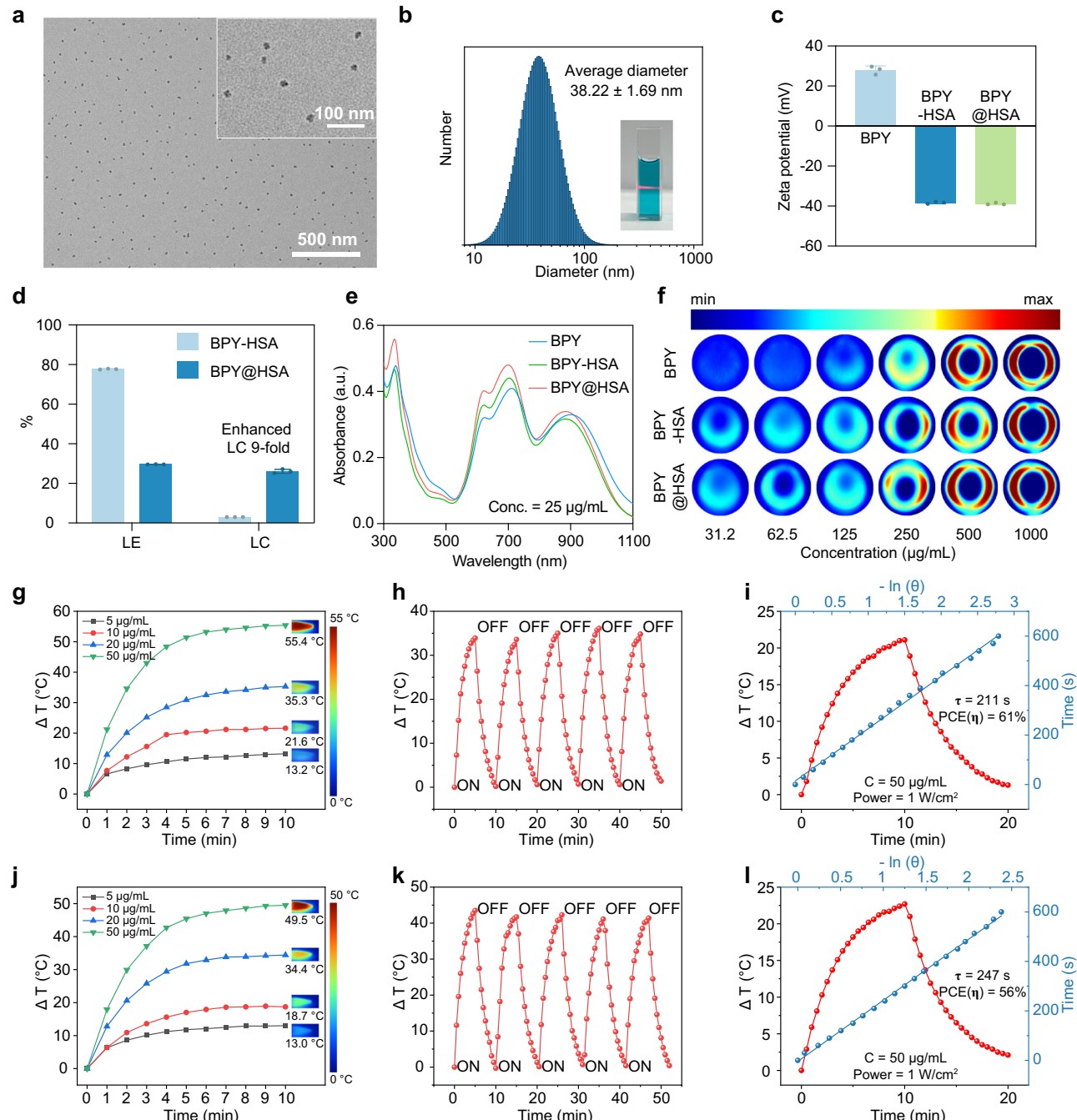

**Fig. 2 | Characterization and photothermal performance of BPY-HSA and BPY@HSA. a** TEM images of BPY@HSA. **b** DLS size distribution of BPY@HSA aqueous solution (insert: photograph of BPY@HSA aqueous solution and observed Tyndall effect). **c** Zeta potentials of BPY, BPY-HSA, BPY@HSA aqueous solutions ($n = 3$ independent experiments). **d** Loading efficiency (LE) and loading capacity (LC) of BPY-HSA and BPY@HSA ($n = 3$ independent experiments). **e** UV-vis-NIR absorption spectra of BPY (in water containing 1% DMSO), BPY-HSA and BPY@HSA aqueous solution. **f** Photoacoustic imaging results of BPY, BPY-HSA, BPY@HSA with

indicated concentrations. Photothermal temperature elevation curve of **g** BPY-HSA and **j** BPY@HSA under 1 W/cm² laser irradiation (wavelength = 808 nm). Photothermal stability of **h** BPY-HSA and **k** BPY@HSA under 1 W/cm² laser irradiation (wavelength = 808 nm). Heating and cooling curves (red curves) and corresponding linear relationships (blue curves) between time and -lnθ from the cooling period for PCE determination of **i** BPY-HSA and **l** BPY@HSA. Data are presented as mean values ± standard deviation (SD). Source data are provided as a Source Data file.

different formulations in Chlorpromazine (inhibits AP-2, which recruits clathrin to cytosolic receptors) treated groups presented significant differences, revealing that the Cy5-BPY@HSA nanoparticle formulations may possess a higher affinity with cytosolic receptors which initiated the CME process, which was the evidence of receptor-mediated endocytosis. In a word, the uptake of the two formulations relied on energy-dependent pathways, and the Cy5-BPY@HSA

nanoparticle formulation presented higher levels on receptor-mediated endocytosis. Moreover, there were some discrepancies in the uptake mechanism of the two formulations between the different cell lines (Supplementary Note 2). For instance, the uptake variation of Cy5-HSA formulation on mβ-CD, Genistein, or Dynasore treated 4T1 and MCF-7 indicates that the MCF-7 relied more on FEME, and 4T1 relied more on caveolin-mediated endocytosis.

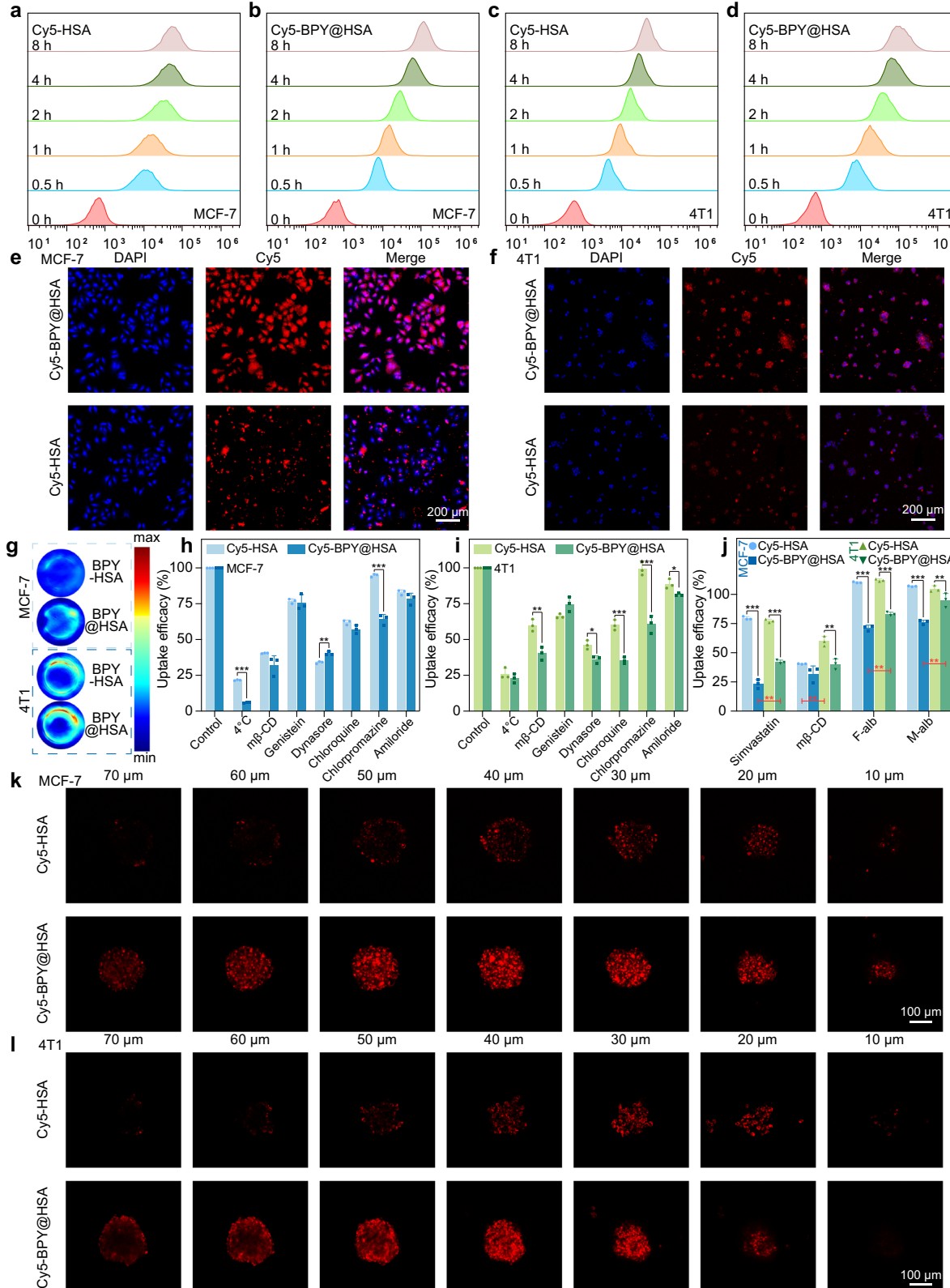

**Fig. 3 | In vitro cellular uptake evaluation and infiltration in 3D tumor spheroids.** Fluorescence histograms of **a, b** MCF-7 and **c, d** 4T1 cells administrated with Cy5-HSA and Cy5-BPY@HSA at different time, respectively. **e** MCF-7 and **f** 4T1 cellular uptake CLSM images (Blue: DAPI channel for nuclei detection; Red: Cy5 channel for Cy5 detection). **g** Photoacoustic imaging of cells treated with BPY-HSA or BPY@HSA for 6 h (1 × 10⁶ cells). Cellular uptake mechanism investigations for **h** MCF-7 and **i** 4T1 cell lines (*n* = 3 biologically independent samples). **j** Receptor

interaction investigations of Cy5-HSA and Cy5-BPY@HSA formulations on MCF-7 and 4T1 cell lines (*n* = 3 biologically independent samples). Z-stack CLSM images of infiltration observation on **k** MCF-7 and **l** 4T1 3D tumor spheroids. Data are presented as mean values ± SD. Statistical differences of *p* values were determined by One-way ANOVA, which were represented as * (*p* < 0.05), ** (*p* < 0.01), *** (*p* < 0.001). Source data are provided as a Source Data file.

The major receptors mediating cellular uptake of the HSA and HSA-based materials are gp60, gp18, and gp30[47]. As the mechanism studies indicated that the receptors may play important roles in the endocytosis of these formulations, we explored the major receptors that interacted with Cy5-HSA and Cy5-BPY@HSA. As shown in Fig. 3j, the Cy5-BPY@HSA formulation presented gp18 and gp30 receptor-depended endocytosis, whereas Cy5-HSA not, and the Cy5-BPY@HSA formulation presented preferred interaction with the gp18 receptor. In addition, there were some variations between the different cell lines as well: the uptake of Cy5-HSA formulation depended on gp60 receptor-mediated endocytosis in MCF-7 and 4T1 cell lines, and Cy5-BPY@HSA formulation depended on gp18 and gp30 receptor-mediated endocytosis in MCF-7 cell line, and it depended more on gp18 and gp60 receptor-mediated endocytosis in 4T1 cell line. To summarize these differences, we demonstrated the uptake pathways and receptor interactions involved in BPY-HSA and BPY@HSA endocytosis through a schematic illustration in Supplementary Fig. 22.

We further evaluated the tumor infiltration performances of Cy5-BPY@HSA and Cy5-HSA on MCF-7 and 4T1 tumor spheroids with different sizes (100, 150, and 200 μm). As shown in Fig. 3k, Cy5-BPY@HSA presented deeper penetration depth than that of Cy5-HSA in both 4T1 and MCF-7 tumor spheroids. Interestingly, we found that Cy5-HSA showed better infiltration ability in 4T1 models than that of MCF-7 (Supplementary Fig. 23). Besides, the Cy5-HSA exhibited size relative infiltration in 4T1 3D tumor spheroids, the smaller size the better infiltration (Fig. 3l). Especially in 100 μm 4T1 3D tumor spheroids, the Cy5-HSA even showed equivalent with Cy5-BPY@HSA (Supplementary Fig. 24). Such infiltration discrepancy on MCF-7 and 4T1 models might attribute to the inherent characteristics of the different cell lines.

## In vitro photothermal anti-tumor performance

Inspired by the good photothermal properties and enhanced cellular uptake of BPY@HSA, we examined the in vitro photothermal anti-tumor performance of BPY@HSA and BPY-HSA. To begin with, we assessed the dark toxicity of BPY-HSA and BPY@HSA on MCF-7, 4T1, and B16 cells (Fig. 4a, b), and the cell viabilities of all groups at a concentration of 500 μg/mL were higher than 80%, indicating both BPY@HSA and BPY-HSA were highly biocompatible without light irradiation. Next, we explored their photothermal cytotoxicity (Fig. 4c, d). In the presence of 808 nm NIR laser, both BPY-HSA and BPY@HSA displayed concentration-relied phototoxicity trend on all the tested cell lines, with low median inhibitory concentration ($IC_{50}$) values, 9.0–9.8 μg/mL for BPY-HSA and 8.4–8.9 μg/mL for BPY@HSA, respectively (Supplementary Table 2). The vibrant in vitro phototoxicity of BPY@HSA against these three cell lines suggested that it might represent a potent anticancer candidate for different kinds of tumors. Live/dead cell staining assays were conducted to visualize the cell survival status after different treatments (Fig. 4e), of which green fluorescence represents live cells and red fluorescence represents dead cells. The fluorescence microscope images indicated that, the MCF-7, 4T1, and B16 cells treated by NIR laser, BPY-HSA and BPY@HSA presented intense green fluorescence, indicating that solely treating with the laser or the formulations cannot influence the cell viabilities. On the other hand, vivid red fluorescence was observed in BPY-HSA + L and BPY@HSA + L groups. These results also confirmed the photothermal-related cytotoxicity of BPY-HSA and BPY@HSA.

In order to further investigate the apoptosis of tumor cells after photothermal treatment, we performed a cell apoptosis study by FCM (Fig. 4f, g, Supplementary Figs. 25–28). Similar with live/dead staining assay results, the BPY-HSA, BPY@HSA, and the negative control group with NIR laser irradiation presented no appreciable apoptotic ratio, whereas the cell apoptosis rates sharply increased by the formulations co-incubation plus laser irradiation. The BPY@HSA + L groups exhibited higher apoptosis rates (95.8% for MCF-7 and 97.3% for 4T1) than that of BPY-HSA + L group (59.9% for MCF-7 and 82.3% for 4T1) in both

cell lines (Supplementary Figs. 26 and 28), which might be the result of the enhanced cellular uptake of BPY@HSA (Fig. 3a–d), because the intracellular PTAs did cause stronger photothermal antitumor effects than that of extracellular PTAs (Supplementary Note 3). The above results suggested that although BPY-HSA and BPY@HSA could both serve as PTAs for tumor PTT in vitro, BPY@HSA outperformed BPY-HSA and showed better therapeutic efficacy.

## Photothermal therapy on xenografted tumor model

Encouraged by the prominent photothermal therapeutic efficacy of BPY-HSA and BPY@HSA in vitro, the therapeutic efficacy was then examined on the xenografted breast cancer (MCF-7) model, followed by the treatment procedure shown in Fig. 5a. When the tumor volumes reached around 100 mm³, the mice were randomly divided into seven groups for different treatments: (I) PBS, (II) PBS + L, (III) BPY, (IV) BPY@HSA, (V) BPY + L, (VI) BPY-HSA + L, (VII) BPY@HSA + L. All groups were administrated by the corresponding formulations on day 1 and 3, and groups treated with laser received 808 nm NIR laser irradiation at 24 h post-administration.

We firstly assessed the tumor accumulation ability of BPY-HSA and BPY@HSA on the MCF-7 model via photoacoustic imaging (Fig. 5b). BPY@HSA exhibited more distinguished photoacoustic signals at tumor regions than that of BPY-HSA, and the photoacoustic intensities in BPY@HSA group gradually increased and peaked at 24 h, indicating the best therapeutic window. The biodistribution of the two formulations (Cy5-labeled Cy5-HSA and Cy5-BPY@HSA formulations) was observed by fluorescence imaging. As shown in Supplementary Fig. 29, the Cy5-BPY@HSA presented enhanced tumor accumulation, whereas Cy5-HSA showed systemic distribution in the whole body, which demonstrated the good tumor target efficacy of Cy5-BPY@HSA (the biodistribution in the body was discussed in Supplementary Note 4). Furthermore, the blood concentration changes against time (Supplementary Fig. 30) were also determined to evaluate the pharmacokinetic property of BPY@HSA, and the half-life of BPY@HSA was calculated as 7.64 h in BALB/c nude mice. Having confirmed that BPY@HSA displayed better tumor accumulation performances, we next investigated the in vivo PTT anti-tumor efficacy on the xenografted breast cancer model. The temperature variations were monitored to study light-to-heat efficacy in vivo during the PTT, as shown in Fig. 5c, d, the BPY and BPY-HSA groups showed no obvious changes compared with the PBS group, but the BPY@HSA group exhibited fast temperature elevation under the laser irradiation, which increased to 52.0 ± 1.9 °C within 10 min. Furthermore, to evaluate the anti-tumor effect, the tumor volumes in all the groups were recorded to day 20 after PTT (Fig. 5e). Among all the tested groups, the BPY@HSA + L group presented the most pronounced therapeutic effect on the MCF-7 model after the PTT treatment. In comparison, tumors in the BPY-HSA + L group showed an uncontrollable and rapid growth profile, which was due to the weak tumor accumulation of BPY-HSA (Supplementary Fig. 31). Besides, the body weight in all groups remained stable during the whole treatment process (Fig. 5f), demonstrating that our treatment procedures featured good compliance with mice. At the end of observation, the tumors were harvested and weighed to further assess the therapeutic effect (Fig. 5g, Supplementary Fig. 32). The remarkable differences on tumor size and weight of the BPY@HSA + L group validated that this bridged BPY@HSA platform could serve as an efficient PTA to suppress tumor growth. Moreover, the tumor burden (Supplementary Fig. 33) of the BPY@HSA + L group was significantly lower than other groups, implying that the BPY@HSA + L treated mice were estimated with a better prognosis.

The harvested tumors were also subjected to further analysis to investigate the intrinsic changes of photothermal treated tumor tissues (Fig. 5h). Compared with other treatments, the BPY@HSA + L group showed the most abnormal cell morphology in hematoxylin and eosin (H&E) staining result, revealing that the tumor tissue was

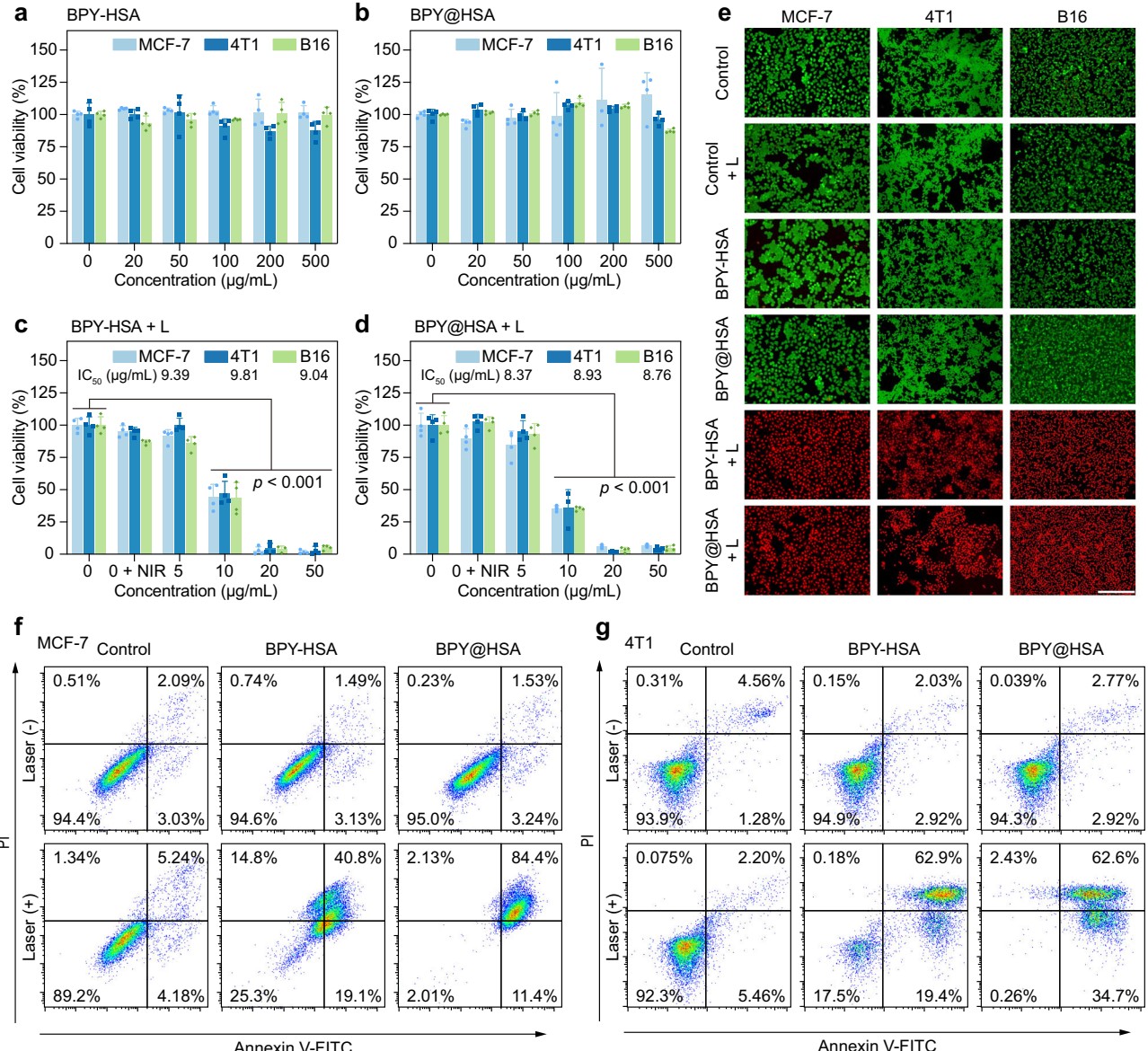

**Fig. 4 | Photothermal therapeutic effect evaluation in vitro. a, b** Dark cyto-toxicity of BPY-HSA and BPY@HSA on MCF-7, 4T1 and B16 cells (*n* = 4 biologically independent samples). **c, d** Cytotoxicity of BPY-HSA and BPY@HSA on different cells under 1 W/cm² laser irradiation (*n* = 4 biologically independent samples). Statistical differences of *p* values were determined by One-way ANOVA.

**e** Fluorescence images of Calcein-AM/PI stained cells with indicated treatments, scale bar = 200 μm. **f, g** Apoptosis assays of MCF-7 and 4T1 cells after indicated treatment. Data are presented as mean values ± SD. Statistical differences of *p* values were determined by One-way ANOVA. Source data are provided as a Source Data file.

severely damaged after the treatment. In addition, terminal deoxynucleotidyl transferase dUTP nick end labeling assay (TUNEL) staining results indicated that the BPY@HSA + L treated tumor underwent the highest degree of cellular apoptosis, which was also accompanied by sharply decreased growth rates implied by Ki67 staining (Supplementary Fig. 34). Considering the only therapeutic component in BPY, BPY@HSA and BPY-HSA was BPY-Mal₂, there was no fundamentally constitutional difference in these formulations. We believe that the improved amount of the accumulated BPY@HSA was the decisive reason to induce localized temperature elevation in tumor tissues, which resulted in the demanded therapeutic efficacy.

The biocompatibility and biosafety of these agents were also comprehensively evaluated in vivo. Considering the negligible body weight loss in each group, the BPY and the two formulations would not affect the health of the mice (Fig. 5f). Meanwhile, the routine blood tests (Supplementary Fig. 35) and blood biochemistry assays

(Supplementary Fig. 36) also confirmed the biosafety of BPY@HSA as well as the other groups, as the parameters were in the normal ranges. Besides, the H&E staining slides of major organs (Supplementary Fig. 37) suggested that there were no obvious lesions during the treatment, which evidenced that the BPY@HSA hardly caused side effects. Moreover, the BPY@HSA inherited the blood compatibility from the HSA, which showed negligible hemolysis, similar with HSA (Supplementary Fig. 38). As a result, both BPY-HSA and BPY@HSA presented desirable biocompatibility and safety, but only BPY@HSA exhibited therapeutic effect on the xenografted tumor model.

**Photothermal therapy on orthotopic tumor model**
The PTT therapeutic efficacy study of BPY@HSA was further extended to an orthotopic 4T1 tumor model, a highly metastatic breast cancer subtype. Both the treatment procedure (Fig. 6a) and the treatment groups were the same as that of the xenografted model. The tumor

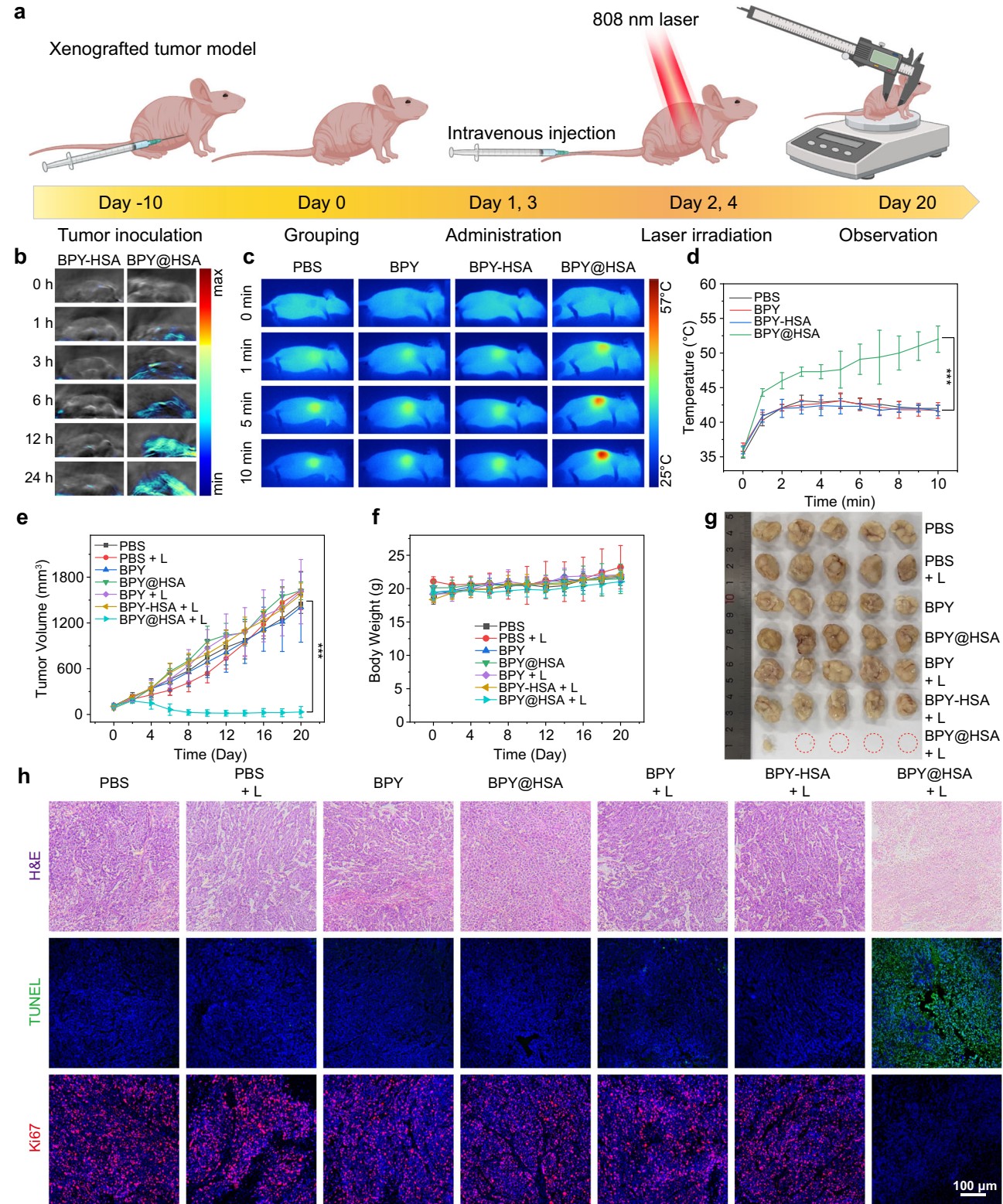

**Fig. 5 | Photothermal therapy on xenografted tumor models. a** Schematic illustration of treatments procedures ($n$ = 5 animals per group). Created with BioRender.com. **b** Photoacoustic imaging results of tumor sites of BPY-HSA and BPY@HSA at different time points ($n$ = 1 animal). **c** Photothermal imaging results of mice administrated with PBS, BPY, BPY-HSA, BPY@HSA after 808 nm laser irradiation (1 W/cm²). **d** In vivo photothermal temperature elevation curves of different groups ($n$ = 5 animals per group). **e** Tumor growth curve of each group ($n$ = 5 animals per group). **f** Body weight of the mice during the treatment ($n$ = 5 animals per group). **g** Digital photo of harvested tumors. **h** Representative slides of H&E staining, TUNEL and Ki67 staining of the tumor tissues after different treatments of each group. Data are presented as mean values ± SD. Statistical differences of $p$ values were determined by One-way ANOVA, which were represented as * ($p < 0.05$), ** ($p < 0.01$), *** ($p < 0.001$). Source data are provided as a Source Data file.

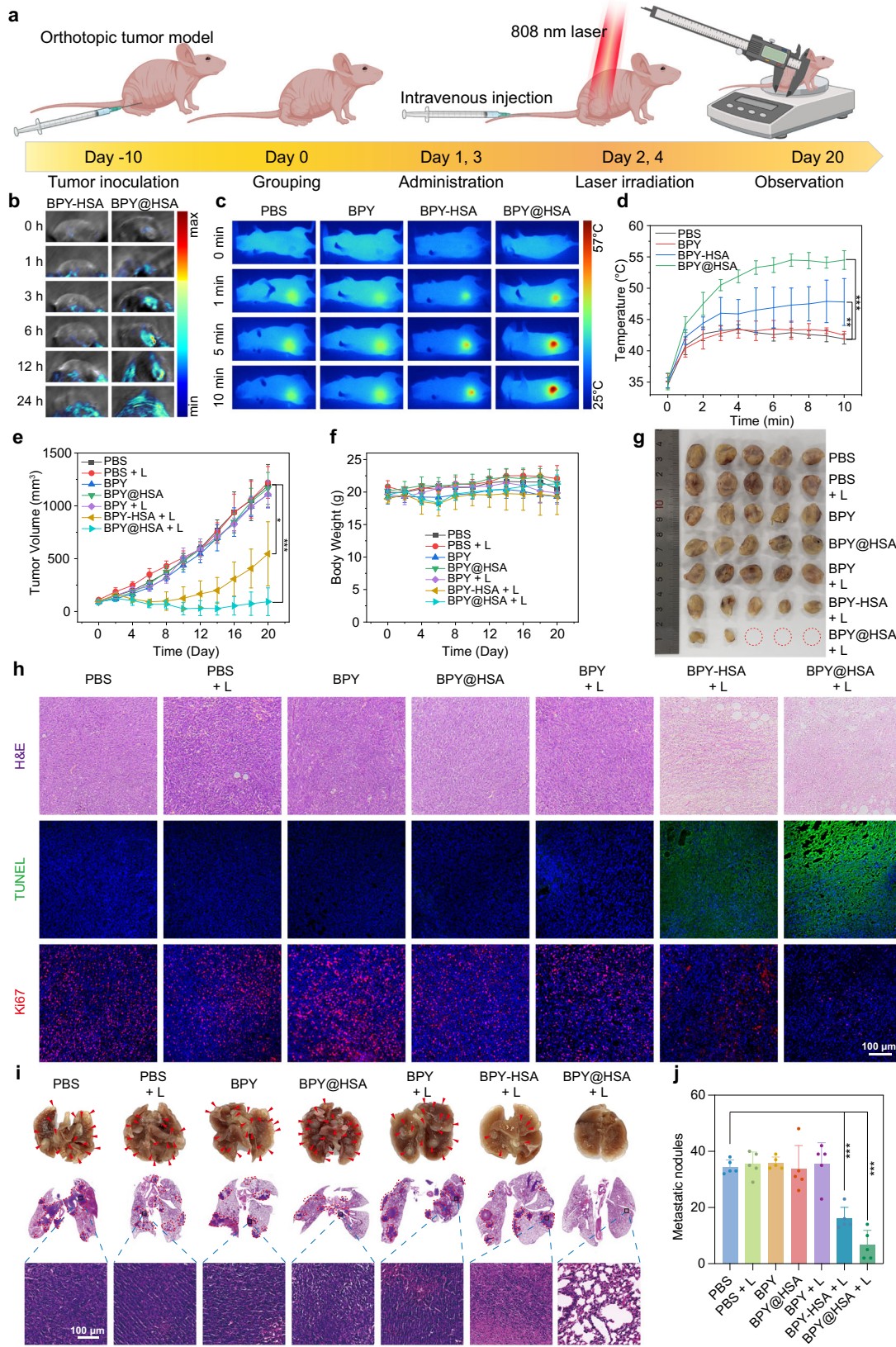

accumulation performances of BPY-HSA and BPY@HSA in 4T1 tumor bearing mice were also evaluated by photoacoustic imaging technology (Fig. 6b). For BPY-HSA, the tumor accumulation in 4T1 tumor was higher than that in MCF-7 tumor, and BPY@HSA exhibited preferential tumor accumulation effect compared with BPY-HSA, with the maximum photoacoustic signal appearing at 24 h post administration. In

addition, the biodistribution of the two formulations (Cy5-labeled Cy5-HSA and Cy5-BPY@HSA formulations) was determined by fluorescence imaging as well. As shown in Supplementary Fig. 39, the Cy5-BPY@HSA presented better tumor accumulation on the 4T1 orthotopic tumor model than that of Cy5-HSA, which only showed a slight accumulation on MCF-7 xenografted tumor model but exhibited

**Fig. 6 | Photothermal therapy on orthotopic tumor models. a** Schematic illustration of treatments procedures (*n* = 5 animals per group). Created with BioRender.com. **b** Photoacoustic imaging results of tumor sites of BPY-HSA and BPY@HSA at different time points (*n* = 1 animal). **c** Photothermal imaging results of mice administrated with PBS, BPY, BPY-HSA, BPY@HSA after 808 nm laser irradiation (1 W/cm$^2$). **d** In vivo photothermal temperature elevation curves of different groups (*n* = 5 animals per group). **e** Tumor growth curve of each group (*n* = 5 animals per group). **f** Body weight of the mice during the treatment (*n* = 5 animals per group). **g** Digital photo of harvested tumors. **h** Representative slides of H&E staining, TUNEL and Ki67 staining of the tumor tissues after different treatments of each group. **i** Representative digital photos and H&E staining slides of harvested lungs. **j** Histogram of metastatic nodules in each group (*n* = 5 animals per group). Data are presented as mean values ± SD. Statistical differences of *p* values were determined by One-way ANOVA, which were represented as * ($p < 0.05$), ** ($p < 0.01$), *** ($p < 0.001$). Source data are provided as a Source Data file.

moderate accumulation on the 4T1 orthotopic tumor model (the biodistribution in the body was discussed in Supplementary Note 5). This type of profile was consistent with the cellular uptake and 3D tumor spheroid infiltration results. We inferred that such a phenomenon might be attributed to the intrinsic diversity of different tumor models, which is also a common phenomenon in the pharmacology and clinical investigations, and we discussed the possible reasons in the Supplementary Note 6.

Next, we examined the light-to-heat conversion efficacy of the two formulations in vivo. As shown in Fig. 6c, d, the BPY-HSA groups exhibited moderate temperature elevation (47.8 ± 3.7 °C), while BPY@HSA showed boosted temperature elevation (54.5 ± 1.5 °C). The photothermal efficacy variation of BPY-HSA on the two tumor models was related to the distinct accumulation behaviors. To further explore the influence of such difference, we recorded the tumor volumes of each group after PTT treatments (Fig. 6e). The tumor volumes of both BPY-HSA + L and BPY@HSA + L groups decreased after PTT treatments at day 1 and 3, indicating that both of them could ablate the tumors by heat (Supplementary Fig. 40). However, the BPY-HSA + L treatment cannot fully inhibit the growth of the tumor. Rapid tumor recurrence occurred at day 8 as a result of incompletely ablation of the tumor. In sharp contrast, the BPY@HSA + L group exhibited a favorable therapeutic effect, and only two of the treated mice occurred tumor recurrence. During the treatment, the body weight of the mice kept relatively steady, indicating that the PTT treatment of BPY@HSA was safe (Fig. 6f). The tumors were harvested at the end of the observation, and the digital photos (Fig. 6g) and the weight (Supplementary Fig. 41) of the harvested tumor clearly revealed that BPY@HSA + L treated group presented the best therapeutic effect, which also could be reflected by the lowest tumor burden in the BPY@HSA + L group (Supplementary Fig. 42). Moreover, the tumors were also subjected to H&E, TUNEL, and Ki67 staining to characterize the photothermal effects on tumor tissues (Fig. 6h, Supplementary Fig. 43). The slides of the tumors revealed that the tumors in BPY@HSA + L group were severely damaged by PTT, which exhibited the highest tumor cell apoptosis degree, and lowest proliferation signal than the other groups. We also collected the lungs to evaluate the anti-metastasis ability against this highly metastatic tumor. The representative digital photos and H&E staining slides were shown in Fig. 6i, and the metastatic nodules of each group (Supplementary Fig. 44) were counted in Fig. 6j. These results demonstrated that the HSA@BPY + L treatments showed the best performance in preventing lung metastasis. Since the employed BALB/c nude mice were immunologically deficient, which cannot induce adaptive immune response, thus the inhibition in lung metastasis was attributed to the ablation of the orthotopic tumors[48]. In addition, the other major organs of each group were also subjected to H&E staining, and there was no apparent damage in these slides (Supplementary Fig. 45).

## Discussion

In this work, we have reported a bridging strategy to prepare the HSA based photothermal therapeutic nano-platform, HSA@BPY. In comparison with other HSA-based drug delivery materials, HSA@BPY presented extra high loading capacity owing to the sufficient utilization of the thiol groups in HSA by our "deconstruction-reconstruction" method. The disulfide bonds in HSA were converted to thiol groups for PTA conjugation and crosslinking to afford the nano-sized materials. The as-prepared BPY@HSA exhibited good water dispersity, narrow size distribution, and well stability. In addition, the BPY@HSA showed high photothermal conversion performance and photoacoustic response, which exhibited prominent photothermal therapeutic effects on several kinds of tumor cell lines and tumor-bearing mice models. Meanwhile, the good biocompatibility of BPY@HSA suggests it is an effective and safe PTA for PTT. In summary, the bridging strategy holds promising potential for developing HSA-based crosslinking platforms with high loading capacity and facile preparation feature, and thus it is favorable for further clinical translation.

## Methods

### Preparation of BPY-HSA and BPY@HSA

BPY-Mal$_2$ cannot directly dissolve in water, thus the BPY formulation was prepared by antisolvent precipitation method. The BPY-Mal$_2$ (2.3 μM in DMSO, 0.2 mL) was gradually diluted by ultra-pure water under sonication to form the BPY formulation.

To prepare the BPY-HSA, BPY-Mal$_2$ was covalently conjugated to free thiol groups on the HSA. In brief, HSA (160 mg) was dispersed in ultra-pure water (8 mL), and next BPY-Mal$_2$ (2.88 μM in DMSO, 2 mL) was gently added to the HSA solution while stirring at 1000 rpm. The mixture was stirred at room temperature for 12 h. After that, the crude product was ultra-filtrated by centrifugal filter device (MWCO = 30 kDa) to obtain the BPY-HSA.

To prepare the BPY@HSA, the disulfide bonds were firstly disrupted by TCEP, and then they were reassembled by BPY-Mal$_2$. In brief, HSA (10 mg) and TCEP (3.75 mg, 10 μmol) were dissolved in ultra-pure water, and the mixture was stirred at 1000 rpm for 30 min. Next, BPY-Mal$_2$ (2.88 μM in DMSO, 2 mL) was gently added (0.3 mL/min) to the reaction mixture, and then the mixture was stirred at room temperature for another 12 h. After that, the crude product was ultra-filtrated by centrifugal filter device (MWCO = 100 kDa) to obtain the BPY@HSA. The concentrations of BPY-HSA and BPY@HSA were represented as equivalent concentration of BPY-Mal$_2$.

To prepare the Cy5-labeled formulations, Sulfo-Cy5-Mal (0.5 mg, 0.6 μmol) and HSA (40 mg) were dispersed in ultra-pure water (10 mL), and then the mixture was stirred at 1000 rpm for 12 h. The Cy5-labeled HSA (Cy5-HSA) was obtained by ultrafiltration (MWCO = 30 kDa). The Cy5-HSA could be disassembled by TCEP, and next it could be reassembled by BPY-Mal$_2$ to afford the Cy5-labeled BPY@HSA (Cy5-BPY@HSA).

### Characterizations

TEM images were obtained from Hitachi HT7800 at 80 kV acceleration voltage. The hydrodynamic diameter of the nanoparticle and zeta-potential were determined by Brookhaven NanoBrook 90Plus PALS (Particle solutions, version 3.6.0.6376). Optical absorption spectra were recorded on PerkinElmer LAMBDA 365. Thermal images and temperature were recorded on a thermal imaging camera (Fluke Ti450). Photoacoustic properties were determined by iThera medical inVision 128 (viewMSOT, version 3.8). An 808 nm laser was employed as the photothermal irradiation source. CLSM images were acquired on a Zeiss LSM880 (ZEN, version 3.0). Flow cytometry was performed on BeckmanCoulter CytoFLEX S (CytExpert, version 2.3) and analyzed by FlowJo (Version 10.8.1). The optical density of 96-well plate was

determined by a plate reader (Epoch 2). A fluorescence microscope (Nikon TS2) was employed to observe the fluorescence and morphology of cells. The fluorescence images were obtained by IVIS Lumina II (Living image, version 4.2). The LE and LC were calculated by following equations:

$$LE(\%) = \frac{Weight_{loadedBPY}}{Weight_{addedBPY}} \times 100\% \qquad (1)$$

$$LC(\%) = \frac{Weight_{loadedBPY}}{Weight_{formulation}} \times 100\% \qquad (2)$$

### Evaluation of photothermal properties
The photothermal properties of BPY, BPY-HSA and BPY@HSA were assessed by irradiating the solutions with an 808 nm laser under different conditions. Real-time thermal images were recorded by a thermal imaging camera to monitor the photothermal properties influenced by laser power density and concentration. The photothermal stability was assessed in a similar way. Five cycles of heating-cooling processes were performed on indicated solutions, and the real-time thermal images were recorded to evaluate the photothermal stability. To determine PCE, 1 mL of aqueous solution was placed in a 1 cm × 1 cm quartz cell, and it was irradiated by 808 nm laser to maximum temperature and then cooled down. The temperature was recorded and the PCE ($\eta$) was calculated by analyzing the heating and cooling curves[24].

### Cell culture
Human breast cancer cells (MCF-7) and murine breast cancer cells (4T1) were cultured in complete DMEM-H medium, and murine melanoma cells (B16) were cultured in complete RPMI-1640 medium. The complete medium was added with 10% FBS and 1% penicillin & streptomycin, and the cells were cultured at 37 °C with an atmosphere containing 5% $CO_2$.

### Cellular uptake
MCF-7 and 4T1 cells ($1 \times 10^5$ per well) were seeded in 6-well plates for 12 h, and the cells were incubated with 5 μg/mL Cy5-HSA or Cy5-BPY@HSA for 0.5, 1, 2, 4 and 8 h, respectively. The cells were washed by PBS and trypsinized for flow cytometry analysis. To observe the cellular uptake, the cells were incubated with Cy5-HSA or Cy5-BPY@HSA for 4 h. After being washed by PBS for 3 times, the cells were stained with Hoechst 33342 for CLSM imaging (excitation: 405 nm for Hoechst 33342, 633 nm for Cy5, respectively). To investigate the cellular uptake effect by photoacoustic imaging, the cells ($1 \times 10^6$ per dish) were incubated with BPY-HSA and BPY@HSA for 6 h, and the cells were washed by PBS and trypsinized for photoacoustic imaging.

To investigate the detailed uptake mechanism of the two formulations, MCF-7 and 4T1 cells ($1 \times 10^5$ per well) were seeded in 6-well plates for 12 h, and the cells were pretreated with 4 °C (affect all energy-dependent pathways), mβ-CD (10 mM, block caveolae and CLIC, also block gp60), Genistein (0.37 mM, inhibit tyrosine kinase), Dynasore (50 μM, block dynamin-dependent pathways), Chloroquine (100 μM, affect clathrin-coated pit-mediated endocytosis, CME), Chlorpromazine (28 μM, inhibit adaptor complex AP2 which mediates CME), and Amiloride (33 mM, inhibit macropinocytosis) to block different uptake pathways for 1.5 h. Then, 5 μg/mL Cy5-HSA and Cy5-BPY@HSA formulations were incubated with the cells for another 4 h. To investigate the proposed receptor interacted with the two formulations, MCF-7 and 4T1 cells ($1 \times 10^5$ per well) were seeded in 6-well plates for 12 h, and the cells were pretreated with Simvastatin (160 μM, block caveolae and CLIC pathways but not block gp60), F-alb (1 mg/mL, competitively inhibit gp18), and M-alb (1 mg/mL, competitively inhibit gp30) for 1.5 h, and 5 μg/mL Cy5-HSA and Cy5-BPY@HSA

formulations were incubated with the cells for another 4 h[47]. The cells were washed with PBS and trypsinized for flow cytometry analysis, and the uptake efficacy was calculated by MFI ratio of indicated pretreatment to the control group.

3D tumor spheroids were established to investigate the deep infiltration ability of the formulations. In brief, $1 \times 10^5$ tumor cells were seeded in 6-well ultra-low adhesion treated plate, and the plate was incubated in complete medium for several days. When the average size of the 3D tumor spheroids reached 100, 150 and 200 μm, the spheroids were incubated with 5 μg/mL of Cy5-HSA or Cy5-BPY@HSA for 12 h. After that, the spheroids were washed by PBS for 3 times and observed by CLSM, and the cross-section images at different depths were scanned by Z-stack mode. The images were further analyzed by ImageJ (Version 1.54c) to examine the penetration ability.

### In vitro cytotoxicity and apoptosis
MCF-7, 4T1, and B16 cells ($5 \times 10^3$ cells per well) were seeded in 96-well plates for 12 h, and then the cells were treated with BPY-HSA or BPY@HSA of indicated concentration. For NIR laser treated group, the incubated cells were irradiated by 808 nm laser (power density = 1 W/$cm^2$) for 10 min at 12 h post administration, and the irradiated cells were cultured for another 12 h. For dark group, the cells were cultured in the dark for 24 h. Cell viability of dark or NIR laser treated groups were assessed using CCK-8 kits. Calcein-AM/PI staining kit was used to visualize live and dead cells of each group, and the fluorescence images were recorded by a fluorescence microscope (excitation wavelength: 470 nm for Calcein-AM, 560 nm for PI).

MCF-7 and 4T1 cells ($1 \times 10^5$ per well) were seeded in 6-well plates for 12 h, and then the cells were treated with 20 μg/mL BPY-HSA or BPY@HSA for 12 h. Next, the NIR laser treated groups were irradiated by 808 nm laser for 10 min. After that, all the groups were cultured for another 4 h and then trypsinized for Annexin V-FITC and PI staining according to the protocol of kit manufacture. The stained cells were dispersed in PBS for flow cytometry assays.

### Ethics statement and animal models
Animal experiment protocols were approved by Animal Ethics Committee of Laboratory Animal Research Center, South China University of Technology (approval number: 2021061), and the animals were raised in SPF grade animal laboratory at the center (maintained at a temperature of ~25 °C in a 40%-70% humidity-controlled environment with a 12 h light/dark cycle).

Female BALB/c nude mice (5 weeks old) were used for establishing tumor models. To establish xenograft subcutaneous breast cancer model, PBS (100 μL) containing $2 \times 10^6$ MCF-7 cells was subcutaneously injected into the right flank of the mice. To establish orthotopic homologous breast cancer model, PBS (100 μL) containing $1 \times 10^6$ 4T1 cells was subcutaneously injected into the fat pat under the fourth mammary gland of the right side. The tumor growth curves were represented by tumor volumes, which were calculated by V = Length × Width$^2$ / 2. During the whole experiment, the maximum tumor volume was limited to 2000 $mm^3$ for each mouse, and the mice were sacrificed when the tumor volume exceeded 2000 $mm^3$.

### Photoacoustic imaging
The photoacoustic images were of sample solutions with indicated concentrations and cell suspensions ($1 \times 10^6$ cells incubated with the formulations for 6 h) were recorded at excitation wavelength of 810 nm. To investigate in vivo tumor accumulation of the formulations by photoacoustic imaging, BPY-HSA or BPY@HSA (0.1 mL, 4 mg/mL) was intravenous injected to a tumor bearing mouse (10 days after tumor implantation) via tail vein, and the photoacoustic images at tumor sites were recorded at 1, 3, 6, 12 and 24 h post-injection.

## Photothermal tumor treatments

The mice with average tumor volume of about 100 mm³ were randomly allocated to 7 treatment groups (biological replicates $n = 5$ for each group), which were PBS, BPY, BPY@HSA, PBS + L, BPY + L, BPY-HSA + L, BPY@HSA + L. The tumor volume and body weight of each mouse was measured every two days to evaluate the tumor treatment efficacy. The dose of each formulation was 20 mg/kg (counted as BPY-Mal₂) for tail vein administration, and the laser irradiation (1 W/cm², 10 min) was performed at 24 h after administration. During the PTT processes, the mice were anesthetized by isoflurane and the temperatures of the irradiated sites were monitored by the thermal imaging camera (auto-capture mode with 30 s intervals). The tumors of represented mice were harvested for H&E staining, TUNEL and Ki67 (Anti-Ki67 Rabbit pAb, Servicebio, GB111499, dilution 1:500; Cy3-conjugated Goat Anti-Rabbit IgG, Servicebio, GB21303, dilution 1:300) analysis to evaluate the tumor inhibition effect 24 h after the first laser treatments (or 48 h after the first administration for groups without laser treatments). At the end of the treatments, the blood samples of each mouse were collected for blood routine analysis and blood biochemistry determination, and the major organs of represented mice were harvested for H&E staining to evaluate the safety of our formulations.

## Statistics and data reproducibility

Quantitative data supporting our study were represented as mean ± standard deviation (SD), and the statistical analysis were conducted using One-way ANOVA for multiple group comparisons (Origin Pro 2021 software) (*: $p < 0.05$, **: $p < 0.01$, ***: $p < 0.001$). The experiments were repeated independently at three times with similar results unless otherwise mentioned, and the micrographs from animal experiments were the representative data from individual samples (the $n$ was given in figure captions). The graphs in the figures were produced by Origin Pro 2021 and Prism 9.4.1, the schemes and illustrations were produced by Adobe Illustrator 2022.

## Reporting summary

Further information on research design is available in the Nature Portfolio Reporting Summary linked to this article.

## Data availability

Source data are provided with this paper. Data that support the findings of this study are available within the Article, Supplementary Information or Source Data file. A reporting summary for this article is also available. Source data are provided with this paper.

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

## Acknowledgements

We greatly acknowledge the financial support from the National Natural Science Foundation of China (32101065 to H. Chen and 82272154 to L. Mei), the start-up fund (BS2021001 to C. Ding) from Maoming People's Hospital, the Guangdong Medical Science and Technology Research Fund (A2021213 to C. Ding), the Fundamental Research Funds for the Central Universities (2021-RC310-005 to L. Mei), the Tianjin Science Fund for Distinguished Young Scholars (22JCJQJC00120 to L. Mei), the Chinese Academy of Medical Sciences Innovation Fund for Medical Sciences (2021-I2M-1-058 and 2022-I2M-2-003 to L. Mei), the Science and Technology Program of Tianjin City (the Basic Research Cooperation Special Foundation of Beijing-Tianjin-Hebei Region, 22JCZXJC00060 to L. Mei), and the National Research Foundation Singapore under Its Competitive Research Programme (NRF-CRP26-2021-0002 to Y. Zhao). Part of elements in the figures were created with biorender.com.

## Author contributions

H.C., L.M. and Y.Z. conceived the project. Z.S., M.L. and Q.H. performed the experiments and analyzed the results. C.D. helped in figure processing. W.W., J.L. and T.C. helped in animal experiments. Y.W., C.L. and X.Z. provided critical information in data processing. Z.S., Q.H. and H.C. wrote the manuscript, and all other authors revised the manuscript.

## Competing interests

The authors declare no competing interests.
