## [Peer Review File · Nature Communications]

NIR-dye bridged human serum albumin reassemblies for effective photothermal therapy of tumorReviewers' comments:

Reviewer #1 (Remarks to the Author):

There has always been an interest in the pharmaceutical application of human serum albumin (HSA) as a drug carrier in academia as well as industry. Recent decades have seen continuing efforts and various researches trying to tap the full potential of HSA. Therefore, I agree with the authors that it is of great importance to develop advanced preparation technologies for constructing HSA-based healthcare materials. In this manuscript, Shi et al. reported a well-designed HSA-based photothermal agent for theranostic purpose. In particular, this study sheds new light on how to develop highly efficient HSA-modification method. They exploited the disulfide bonds inside HSA by firstly reducing them into thiol groups and then using bi-maleimide functionalized photothermal agent to react with thiol groups and simultaneously crosslinked the disrupted HSA segments back together to afford the well-defined nano-structure. Some of the current major bottlenecks of HSA-based nanomaterials were solved by this systems. The drug loading capacity was improved and the preparation process was simplified. According to the experimental data in this manuscript, the reported BPY@HSA exhibited good optoacoustic performance as well as satisfying photothermal tumor inhibition efficiency. Generally speaking, this is an interesting study and the reported method provides a new perspective for development of advanced HSA-based materials. I believe this manuscript could inspire researchers in this field to study more about HSA-based nanomaterials. There are only a few points that need to be addressed.

1. In FIG. S33, I do not see the unit of the tumor volume in the left axis title.
2. I do not know what "MCG" means in FIG. S29. Maybe it should be correct as "MCH". The authors should carefully check all the abbreviation in this figure.
3. I think there was a mistake in FIG 2i, the title of the left axis should be corrected as Delta T (ΔT). The authors should check on this. And this also goes for Fig 2l.
4. The authors should pay attention to the description of some results. There are several experiments that need to be stated more clearly. For example, in the caption of FIG 2, the authors only present the irradiation intensity, they missed out the wavelength of the external light. Also in Fig S32, I do not see what the numbers in this figure means, the unit was also missed.
5. I suggest the authors should supplement some experiment details for readability, such as the TUNEL and Ki67 staining tumor section results. I wonder when exactly did the tumor tissue sample were detected? The day after irradiation or at the end of the whole experiment? The authors should clarify this point.

Reviewer #2 (Remarks to the Author):

This work describes a Human Serum Albumin (HSA) nanoparticle formed from HSA polypeptide chains cross-linked with a photothermal active BODIPY (BPY) dye for cancer therapy. The approach is based on reduction of HSA into polypeptides to allow availability of multiple free thiols for maleimide conjugation of a bi-maleimide functionalised BODIPY (BPY-MaI2). The authors evaluate physicochemical properties and dye loading capacity of the BPY@HSA compared to a non-reduced HSA-BODIPY conjugate (BPY-HSA), and investigate in vitro and in vivo the delivery efficacy and photothermal therapeutic properties in a xenograft and orthotopic tumour models in mice. The work is interesting in the field of albumin-based drug delivery and photothermal therapy, however, a number of issues need to be addressed.

1. The authors suggest that the system can be applied for the delivery of other drug types in addition to a photothermal agent, stating in the discussion "The methodology developed here also may be used to incorporate more bi-maleimide functionalized therapeutic agents into HSA-based platforms." A limitation of this system is the requirement for release of certain drug types that need to dissociate from the albumin that may require stimuli-responsive linkers complicating the design. Could the authors comment?
2. The cross-linker interactions should be clearly described. Figure 1, a schematic seem to show

cross-linking of 2 polypeptide chains by the BPY-Mal, but what drives the assembly of multiple polypeptide chains required to form the nanoparticle. How many polypeptide chains are in the nanoparticle? How can the cross-linking be controlled to form discrete nanoparticles rather than aggregates.

The authors state in line 129 "When BPY@HSA was incubated in water or in DMEM complete medium (containing 10% of FBS), the size of BPY@HSA showed negligible changes for up to 7 days (Supplementary Fig. 16), indicating the good stability of BPY@HSA, which is important for further in vivo application." The nanoparticles are almost double in size in DMEM and 10 % FBS compared to in water. Does this indicate instability?

3. The authors cite in Supplementary Table 1, HSA systems based on cross-linked albumin fragments and cysteine cross-linking. What is the novelty of the authors approach compared to these systems? A discussion should be included in the manuscript.

4. The authors cite in line 48 "This strategy brings benefits such as enhanced tumor uptake efficiency mediated by gp60 receptor and enhanced permeability and retention (EPR) effect." Also line 161, "Owing to overexpressed albumin-binding proteins in tumor cells and the nutrient delivery function of HSA, HSA-based therapeutic platforms have unique advantage for tumor uptake". Would undefined assembly of polypeptide chains into a nanoparticle not compromise the inherent structure of native albumin and its domains needed for its molecular interactions? It would be relevant to investigate the ability of the nanoparticle compared to native full structured albumin to interact with cellular receptors. This is relevant if the authors propose active targeting by the system.

5. The authors compare the BPY@HSA nanoparticle system to a BPY-HSA conjugate. Would differences in size between these two systems influence in vitro and in vivo delivery? This should be mentioned in the work.

6. Figure 2, LE and LC abbreviations in the figure caption should be also be written in full. What does the insert in 2b mean to show?

7. In the in vitro delivery work, what cellular receptors do the authors suggest albumin interacts with?

Figure 5, the BPY@HSA and BPY-HSA seemed to elicit similar cytotoxicity in the presence of a NIR laser (Figure 3 c and d), but the authors state "BPY@HSA+L group exhibited higher apoptosis rates than that of BPY-HSA+L group in both MCF-7 and 4T1 cells, which might be the result of the enhanced cellular uptake of BPY@HSA." Can the authors comment and include numbers in the text to support the statement.

8. Albumin is used as a drug half-life technology, it would be relevant to show the half-life for the BPY@HSA nanoparticle system.

9. The number of animals are not mentioned in the figures 5 and 6. This is important to include. In the supplementary 5 animals is cited, is this the number used in figure 5 and 6.

10. Why did BPY-HSA have a therapeutic effect in the orthotopic model (Figure 6) but not the xenograft model (Figure 5)?

11. Photoacoustic should be described in the introduction. The types of albumin drug delivery approaches e.g. non-covalent association, conjugation, nanoparticles, should be more clearly explained in the introduction.

12. Figure 1, the BPY-Mal2 molecule should be clearly defined, does purple depict the BPY, and what are the 4 strands from this when it has only 2 maleimide groups.

13. In the Material and Methods, what stain was used in TEM, what medium were the nanoparticles in for zeta potential and size measurements? Cell numbers should be stated for the in vitro work, animal numbers should be included and instrumentation for PA and PT imaging mentioned.

14. The grammar should be addressed in the manuscript.

Reviewer #3 (Remarks to the Author):

In this manuscript, the authors reported a bridging strategy to fabricate a novel HSA-based photothermal agent nano-platform. The as-prepared BPY@HSA presented better PTT activity and photoacoustic response than BPY-HSA, which was obtained by modifying the pristine HSA with BPY-Mal2 directly. The BPY@HSA could effectively accumulate in tumor sites and serve as a safe PTA to photothermally ablate the tumors. Nevertheless, the article is interesting with a wide range of studies, which deserve publication, but in my opinion, not of outstanding significance for publication in Nature Communications.

1. The absorption spectrum of BPY in Fig. 2e was obtained in water containing 1% DMSO. Does it aggregate in the solution. Are the absorption spectra of BPY (in water containing 1% DMSO), BPY-HSA and BPY@HSA at the same concentration. Why are they different.
2. The photothermal conversion efficiencies of BPY-HSA and BPY@HSA are so high, why not a lower laser power density is used. The permissible laser power density of 808 nm laser is far below 1 W/cm².
3. Laser irradiation was applied twice, which is not common for tumor photothermal therapy. For most works, administration and laser irradiation are performed only once, you should explain about the operation.
4. Why did Cy5-BPY@HSA present deeper penetration depth than that of Cy5-HSA in both 4T1 and MCF-7 tumor spheroids. The surface of Cy5-BPY@HSA and Cy5-HAS should be similar.
5. The manuscript was not carefully prepared. Many letters in the figures are obscured.

Response to Reviewers' Comments

Reviewer #1

There has always been an interest in the pharmaceutical application of human serum albumin (HSA) as a drug carrier in academia as well as industry. Recent decades have seen continuing efforts and various researches trying to tap the full potential of HSA. Therefore, I agree with the authors that it is of great importance to develop advanced preparation technologies for constructing HSA-based healthcare materials. In this manuscript, Shi et al. reported a well designed HSA-based photothermal agent for theragnostic purpose. In particular, this study sheds new light on how to develop highly efficient HSA-modification method. They exploited the disulfide bonds inside HSA by firstly reducing them into thiol groups and then using bi-maleimide functionalized photothermal agent to react with thiol groups and simultaneously crosslinked the disrupted HSA segments back together to afford the well-defined nanostructure. Some of the current major bottlenecks of HSA-based nanomaterials were solved by this systems. The drug loading capacity was improved and the preparation process was simplified. According to the experimental data in this manuscript, the reported BPY@HSA exhibited good optoacoustic performance as well as satisfying photothermal tumor inhibition efficiency. Generally speaking, this is an interesting study and the reported method provides a new perspective for development of advanced HSA-based materials. I believe this manuscript could inspire researchers in this field to study more about HSA-based nanomaterials. There are only a few points that need to be addressed.

Response: We appreciate your positive feedback and constructive suggestions to improve our work. We are also grateful for this opportunity to revise these problems in the manuscript.

1. In FIG. S33, I do not see the unit of the tumor volume in the left axis title.

Response: Thanks for pointing out this issue, and we have added the unit of tumor volume in the left axis title in the revised form.

Modification:

Supplementary Fig. 31. Tumor growth curves of MCF-7 tumor model ($n = 5$) with different treatments indicated.

Supplementary Fig. 40. Tumor growth curves of 4T1 tumor model ($n = 5$) with different treatments indicated.

2. I do not know what “MCG” means in FIG. S29. Maybe it should be correct as “MCH”. The authors should carefully check all the abbreviation in this figure.

Response: Thanks for pointing out this issue, and we have checked all abbreviation in our manuscript word by word.

Modification:

Supplementary Fig. 35. Routine blood analysis of different treatment groups ($n = 3$).

3. I think there was a mistake in FIG 2i, the title of the left axis should be corrected as Delta T (ΔT). The authors should check on this. And this also goes for Fig 2l.

Response: Thanks for the comments. The typos have been corrected. We have also corrected the temperature as ΔT in Fig. 2i and Fig. 2l.

Modification:

Fig. 2 Characterization and photothermal performance of BPY-HSA and BPY@HSA. **a** TEM images of BPY@HSA. **b** DLS size distribution of BPY@HSA aqueous solution (insert: photograph of BPY@HSA aqueous solution and observed Tyndall effect). **c** Zeta potentials of BPY, BPY-HSA, BPY@HSA aqueous solution. **d** Loading efficiency (LE) and loading capacity (LC) determination of BPY-HSA and BPY@HSA. **e** UV-vis-NIR absorption spectra of BPY (in water containing 1% DMSO), BPY-HSA and BPY@HSA aqueous solution. **f** Photoacoustic imaging results of BPY, BPY-HSA, BPY@HSA with indicated concentrations. Photothermal temperature elevation curve of **g** BPY-HSA and **j** BPY@HSA under 1 W/cm² laser irradiation (wavelength = 808 nm). Photothermal stability of **h** BPY-HSA and **k** BPY@HSA under 1 W/cm² laser irradiation (wavelength = 808 nm). PCE

determination of i BPY-HSA and I BPY@HSA.

4. The authors should pay attention to the description of some results. There are several experiments that need to be stated more clearly. For example, in the caption of FIG 2, the authors only present the irradiation intensity, they missed out the wavelength of the external light. Also in Fig S32, I do not see what the numbers in this figure means, the unit was also missed.

Response: Thank you for your suggestion, and we have addressed these issues and double checked all the figure caption to make them clear.

Modification:

Supplementary Fig. 38. Hemolysis tests for HSA and BPY@HSA. Ctrl. represents negative control group, Posi. represents positive control group, and the concentration of HSA or BPY@HSA is ranged from 3.9 to 250 µg/mL. The hemolysis rates of all tested groups were below 1%, indicating the good biosafety of the BPY@HSA nanoplatfrom.

5. I suggest the authors should supplement some experiment details for readability, such as the TUNEL and Ki67 staining tumor section results. I wonder when exactly did the tumor tissue sample were detected? The day after irradiation or at the end of the whole experiment? The authors should clarify this point.

Response: We thank the reviewer's kind advice on the experiment details. The tumor tissues used

in TUNEL and Ki67 staining were collected 24 h after the first laser treatments (or 48 h after the first administration for groups without laser treatments), which is a commonly employed protocol for evaluating the photothermal therapy effects on tumor tissues. We have appended these details in the method section, and also checked all the method section to clarify this point in the revised manuscript.

Modification:

Photothermal tumor treatments.

The mice with average tumor volume of about 100 mm³ were randomly allocated to 7 treatment groups (biological replicates $n = 5$ for each group), which were PBS, BPY, BPY@HSA, PBS+L, BPY+L, BPY-HSA+L, BPY@HSA+L. The tumor volume and body weight of each mouse was measured every two days to evaluate the tumor treatment efficacy. The dose of each formulation was 20 mg/kg (counted as BPY-Mal₂) for tail vein administration, and the laser irradiation (1 W/cm², 10 min) was performed at 24 h after administration. During the PTT processes, the mice were anesthetized by isoflurane and the temperatures of the irradiated sites were monitored by the thermal imaging camera (auto-capture mode with 30 s intervals). The tumors of represented mice were harvested for H&E staining, TUNEL and Ki67 analysis to evaluate the tumor inhibition effect 24 h after the first laser treatments (or 48 h after the first administration for groups without laser treatments). At the end of the treatments, the blood samples of each mouse were collected for blood routine analysis and blood biochemistry determination, and the major organs of represented mice were harvested for H&E staining to evaluate the safety of our formulations.

Reviewer #2

This work describes a Human Serum Albumin (HSA) nanoparticle formed from HSA polypeptide chains cross-linked with a photothermal active BODIPY (BPY) dye for cancer therapy. The approach is based on reduction of HSA into polypeptides to allow availability of multiple free thiols for maleimide conjugation of a bi-maleimide functionalised BODIPY (BPY-Mal2). The authors evaluate physicochemical properties and dye loading capacity of the BPY@HSA compared to a non-reduced HSA-BODIPY conjugate (BPY-HSA), and investigate in vitro and in vivo the delivery efficacy and photothermal therapeutic properties in a xenograft and orthotopic tumour models in mice. The work is interesting in the field of albumin-based drug delivery and photothermal therapy, however, a number of issues need to be addressed.

Response: Thank you for your positive feedback regarding our paper. We have taken your suggestions into consideration and have made several revisions to the manuscript, including additional experiments to address your concerns. Please find below our point-by-point response to your comments.

1. The authors suggest that the system can be applied for the delivery of other drug types in addition to a photothermal agent, stating in the discussion “The methodology developed here also may be used to incorporate more bi-maleimide functionalized therapeutic agents into HSA-based platforms.” A limitation of this system is the requirement for release of certain drug types that need to dissociate from the albumin that may require stimuli-responsive linkers complicating the design. Could the authors comment?

Response: We appreciate the reviewer’s comment about this point. Indeed, stimuli-responsive linkers are the prerequisite for synthesizing activable prodrugs and realizing controlled drug release purpose. It is inevitable to use this kind of linkers during the synthesis procedure. Our design philosophy is mainly focused on developing a new method to prepare HSA-based formulations in a facile and efficient manner. Based on our strategy, the bi-maleimide structured active pharmaceutical ingredient can be easily loaded into untangled HSA molecules and simultaneously crosslinks them back together to get the nano-structured formulation. As a result, the drug loading and crosslinking steps are combined into one single step.

Up to now, there have been significant advances in the field of stimuli-activable prodrugs (e.g., [10.1039/d2sc01003h](https://doi.org/10.1039/d2sc01003h) and [10.1021/acsnano.2c05379](https://doi.org/10.1021/acsnano.2c05379)). Many research teams have successfully prepared and reported prodrug systems with controllable activation functions by introducing stimuli-responsive linkers into drug molecules. Thus, this technology is relatively mature. Although such a design may involve more synthetic steps, considering the potential advantages, such as precise control of drug release, reduced toxicity, and improved treatment efficacy, this approach

is worthwhile.

2. The cross-linker interactions should be clearly described. Figure 1, a schematic seem to show cross-linking of 2 polypeptide chains by the BPY-Mal, but what drives the assembly of multiple polypeptide chains required to form the nanoparticle. How many polypeptide chains are in the nanoparticle? How can the cross-linking be controlled to form discrete nanoparticles rather than aggregates.

The authors state in line 129 “When BPY@HSA was incubated in water or in DMEM complete medium (containing 10% of FBS), the size of BPY@HSA showed negligible changes for up to 7 days (Supplementary Fig. 16), indicating the good stability of BPY@HSA, which is important for further in vivo application.” The nanoparticles are almost double in size in DMEM and 10 % FBS compared to in water. Does this indicate instability?

Response: Thank you for raising these questions. We would like to explain these concerns in the following four aspects.

(2-1) To prepare the BPY@HSA formulations, the disulfide bonds in HSA were first reduced to thiol groups by the reducing agent TCEP. The obtained thiols were further reacted with bi-maleimide structured BPY-Mal₂ through Michael addition reaction (denoted in red equation in Supplementary Fig. 13). During this process, the BPY-Mal₂ not only acted as photothermal agent for therapeutic purpose, but also crosslinked the cleaved polypeptide segments back together to afford nano-structured BPY@HSA. The hydrophilic polypeptide chains were linked by hydrophobic BPY-Mal₂ molecules, leading to the structure of hydrophilic polypeptide shell and hydrophobic BPY core. Therefore, the joint role of hydrophilic/hydrophobic balance and chemical crosslinking mechanism drives the formation of nanoparticles.

Supplementary Fig. 13. Comparison of BPY-HSA and BPY@HSA formulations. For BPY-HSA, each HSA protein owned only a free thiol group, and thus it could only afford one BPY-Mal₂ molecule. For BPY@HSA, all the disulfide bonds were deconstructed and then reconstructed by BPY-Mal₂ to form BPY-Mal₂-bridged nanoparticles with higher loading capacity.

(2-2) We calculated the ratio between polypeptide chains and loaded drugs according to the loading mechanism and loading capacity of the BPY@HSA. Each polypeptide chain was capable for bridging 6.5 to 7 (6.8 ± 0.35) BPY-Mal₂ molecules in the BPY@HSA nanoparticles. Constrained by the current technology and circumstances, the exact number of the polypeptide chains from each nanoparticle cannot be directly determined. Nevertheless, we roughly estimated the number of polypeptide chains by determining the mass of the nanoparticles and counting the number the nanoparticles by flow nano-analyzing technology. According to our determination, 15.5 $\mu\text{g}/\text{mL}$ of BPY@HSA solution contained 1.79×10^9 nanoparticles, thus there were 4.4×10^5 peptide chains (average) for each nanoparticle, and each nanoparticle contained 3.0×10^6 BPY molecules, which were calculated by molecular weight and Avogadro's constant.

1. Calculating the ratio of BPY and HSA chains

	Content (%)	Mw	n	2n (two chains of a HSA)	Ratio (BPY:HSA)	Ratio (2n, BPY:chain)
HSA	74.8	67000	0.00112	0.00223	13.0	6.5
BPY	25.2	1740	0.01451	-		
HSA	74.1	67000	0.00111	0.00221	13.5	6.7
BPY	25.9	1740	0.01491	-		
HSA	72.8	67000	0.00109	0.00217	14.4	7.2
BPY	27.2	1740	0.01563	-		

2. Counting particle numbers by NanoFCM

Total Concentration Information

	Particle Number	Dilution Factor
STD	6529	100
Blank	39	—
Sample	11186	5
STD Con.	2.10E+10	Particles/mL
Sample Flow Rate	31.09	nL/min
Sample Con.	1.79E+9	Particles/mL
Corrected Ratio:	11147/11147	100.0%

BPY concentration = 15.5 µg/mL

3. Calculating the proposed polypeptide chains in each nanoparticle

BPY@HSA concentration
15.5 µg/mL (count as BPY)

1.79×10^9 BPY@HSA nanoparticles

Mass (g)	Mw	n	NA	Number of molecules
2.56E-14	67000	3.825E-19	6.02E+23	460679
8.66E-15	1740	4.975E-18		2995929
2.47E-14	67000	3.6879E-19		444171
8.66E-15	1740	4.975E-18		2995929
2.32E-14	67000	3.4595E-19		416658
8.66E-15	1740	4.975E-18		2995929

Fig. R1. Calculation process of determining the number of polypeptide chains in nanoparticles.

(2-3) In recent studies, other substances have also been used as chemical crosslinkers to prepare albumin nanoparticles. For example, Patel et al. developed a method for manufacturing albumin nanoparticles loaded with the antibacterial drug benzothiazinone by first incubating HSA with benzothiazinone and then adding glutaraldehyde to crosslinking them into nanoparticles (10.1016/j.jconrel.2020.08.022). Yang et al. synthesized a hypoxia-sensitive azobenzene-based linker molecule to crosslink HSA into nanoparticles (10.1002/adma.201901513). To minimize the formation of aggregates during the crosslinking process, it is crucial to precisely control the feeding ratio of the chemical crosslinkers. We also explored different feeding ratios of BPY-Mal₂ during the preparation process and found discovered that the current feeding ratio is optimal in preventing the formation of aggregates.

In addition, the stirring processes and the reaction temperature were also carefully controlled to avoid locally high concentration, high temperature and heterogeneous phase formation.

(2-4) Considering there is no doubt that almost all the nanoparticles would interact with proteins after intravenous injection, we employed DMEM containing 10 % FBS to investigate whether the interactions between the BPY@HSA nanoparticles and proteins in the blood circulation would affect the stability of BPY@HSA. As exhibited in Supplementary Fig. 16, the particle size of BPY@HSA increased about 40 nm in DMEM containing 10 % FBS, indicating the protein absorption

of the BPY@HSA nanoparticles. This phenomenon is relating to protein corona that is referring to the structure composed of one or more layers of proteins adsorbed on the surface of nanoparticles after entering serum-containing media or body fluids. Besides, the particle size of BPY@HSA in DMEM containing 10 % FBS kept stable in 7 days revealed that such nanoplatform could also keep stable when entering to the blood stream.

3. The authors cite in Supplementary Table 1, HSA systems based on cross-linked albumin fragments and cysteine cross-linking. What is the novelty of the authors approach compared to these systems? A discussion should be included in the manuscript.

Response: We appreciate the reviewer's suggestion. The drug loading strategies of these reported systems in Supplementary Table 1 were different from our BPY@HSA crosslinked systems. The crosslinked systems in Supplementary Table 1 are classic crosslinked systems, which contains three major components: the scaffold HSA molecules, crosslinking agents, and drug molecules. The crosslinking agents served as a glue to bind the HSA molecules together, and the drug molecules were loaded simultaneously by physical inclusion. In our BPY@HSA systems, BPY-Mal₂ not only serves as the crosslinking agents, but also acts as the therapeutic agent of the systems, since the therapeutic agent loading and the crosslinking steps were merged. Therefore, the advantages of our approach are the simplified preparation process, and the therapeutic agent is covalently loaded in the nanoparticles rather than physical inclusion. We have supplemented this discussion in Supplementary Note 1 in the supplementary materials.

Modification:

Notably, the LC of BPY@HSA (26.1%) was 9-times higher than that of BPY-HSA (3.0%), which was also higher than most of the reported HSA-based platforms by orders of magnitude (Supplementary Table 1). AS compared with those formulation strategies, BPY-Mal₂ in our strategy not only serves as the crosslinking agents, but also acts as the therapeutic agent of the systems, since the therapeutic agent loading and the crosslinking steps were merged. Therefore, our methodology offers new avenues for formulating HSA-based therapeutic platforms, which are both feasible and straightforward.

4. The authors cite in line 48 "This strategy brings benefits such as enhanced tumor uptake efficiency mediated by gp60 receptor and enhanced permeability and retention (EPR) effect." Also line 161, "Owing to overexpressed albumin-binding proteins in tumor cells and the nutrient delivery function of HSA, HSA-based therapeutic platforms have unique advantage for tumor uptake". Would undefined assembly of polypeptide chains into a nanoparticle not compromise the inherent structure of native albumin and its domains needed for its molecular interactions? It would be relevant to investigate the ability of the nanoparticle compared to native full structured

albumin to interact with cellular receptors. This is relevant if the authors propose active targeting by the system.

Response: We appreciate the reviewer for raising this question. Firstly, the *in vitro* cellular uptake performances of Cy5-HSA and Cy5-BPY@HSA are different. As shown in Fig. 3a-d, the fluorescence signals of Cy5 became stronger as the incubation time was prolonged, but the peaks in the histograms of different formulations exhibited obvious variations. According to further quantification analysis (Supplementary Fig. 19), the mean fluorescence intensity (MFI) in Cy5-BPY@HSA groups were higher than the Cy5-HSA groups, which indicated that BPY@HSA exhibited better cellular uptake efficiency, and they might undergo different uptake mechanisms when entering tumor cells.

Supplementary Fig. 19. Mean Cy5 intensity of cellular uptake studies for Cy5-HSA and Cy5-BPY@HSA formulations on **a** MCF-7 and **b** 4T1 cell lines.

As suggested, we investigated the uptake mechanism of natural HSA and our bridged nanoparticles Cy5-BPY@HSA to explore the reasons why Cy5-BPY@HSA exhibited superior cellular uptake efficiency. To investigate the detailed uptake mechanism of the two formulations, MCF-7 and 4T1 cells were pretreated with a series of representative endocytosis inhibitors, which have been recently discussed in a review article (10.1038/s41565-021-00858-8).

They include: 4°C (affecting all energy-dependent pathways), mβ-CD (10 mM, for blocking caveolae and clathrin-independent carrier (CLIC) pathways, also blocking gp60), Genistein (0.37 mM, for inhibiting tyrosine kinase), Dynasore (50 μM, for blocking dynamin-dependent pathways), Chloroquine (100 μM, for suppressing clathrin-coated pit-mediated endocytosis (CME)), Chlorpromazine (28 μM, for inhibiting AP2 which mediates CME), Amiloride (33 mM, for inhibiting macropinocytosis) and then incubated with Cy5-HSA and Cy5-BPY@HSA.

To investigate the proposed receptor interacted with the two formulations, MCF-7 and 4T1 cells were also pretreated with Simvastatin (160 μM, for blocking caveolae and CLIC pathways but

not blocking gp60), F-alb (formaldehyde-treated albumin, 1 mg/mL, competitively inhibit gp18), and M-alb (maleylated albumin, 1 mg/mL, competitively inhibit gp30) for 1.5 h, and 5 µg/mL Cy5-HSA and Cy5-BPY@HSA formulations were incubated with the cells for another 4 h (10.1021/acsami.1c03065). The cells were washed with PBS and trypsinized for flow cytometry analysis, and the uptake efficacy was calculated by MFI ratio of indicated pretreatment to the control group.

The cellular uptake mechanism results were shown in **Fig. 3e-f and Supplementary Fig. 20-21**. The uptake of both formulations was greatly involved in energy-dependent pathways as 4 °C treated cells presented the lowest uptake, indicating active transportation processes on the uptake of Cy5-HSA and Cy5-BPY@HSA formulations. In detail, significantly decreased uptake efficacy on mβ-CD groups suggested that gp60, caveolae and CLIC were the essential receptor/pathways affecting the uptake of the two formulations. On the one hand, Cy5-HSA formulation was slightly inhibited by simvastatin (Fig. 3g) but greatly inhibited by mβ-CD because HSA is known for gp60 receptor-mediated endocytosis. On the other hand, Cy5-BPY@HSA formulation presented great inhibition by simvastatin revealing that the caveolae (4T1 only, there is no molecular machinery of caveolae in MCF-7) and clathrin-independent carriers (MCF-7 and 4T1) could act as powerful pathways of the Cy5-BPY@HSA formulations in cellular uptake. The dynamin-dependent endocytosis also served as an efficient pathway in cell uptake, as clathrin-independent/dynamin-dependent endocytosis (FEME) was evidenced by the decrease in Genistein and Dynasore treated groups, and clathrin-mediated endocytosis (CME) was evidenced by the decrease in Dynasore and Chloroquine treated groups. Moreover, Amiloride treated groups demonstrated that both of the formulations were involved in macropinocytosis, and Cy5-BPY@HSA nanoparticle formulation relied more on macropinocytosis. Interestingly, the uptake efficacy on different formulations in Chlorpromazine (inhibits the adaptor complex, AP-2, which recruits clathrin to cytosolic receptors) treated groups presented significant differences, revealing that the Cy5-BPY@HSA nanoparticle formulation may possess a higher affinity with cytosolic receptors which initiate the CME process, which is the evidence of receptor-mediated endocytosis. **In short, the uptake of the two formulations relied on energy-dependent pathways, and the Cy5-BPY@HSA nanoparticle formulation presented higher levels on receptor-mediated endocytosis although it present some discrepancies in the uptake mechanism on the different cell lines.**

The major receptors mediating cellular uptake of the HSA and HSA-based materials are gp60, gp18, and gp30. We further investigated the major receptors that interact with the Cy5-HSA and Cy5-BPY@HSA by simvastatin (block caveolae and clathrin-independent carrier pathways but not block gp60, a compensation of mβ-CD), F-alb (formaldehyde-treated albumin, competitively inhibit gp18), and M-alb (maleylated albumin, competitively inhibit gp30), and F-alb and M-alb

were prepared by previous reported methods (10.1021/acsami.1c03065). As shown in **Fig. 3g**, the Cy5-BPY@HSA formulation presented gp18 and gp30 receptor-dependent endocytosis whereas Cy5-HSA not, and the Cy5-BPY@HSA formulation presented preferred interaction with gp18 receptor. Meanwhile, as HSA is known for its gp60 receptor-mediated endocytosis, the level of gp60 receptor-mediated endocytosis to Cy5-BPY@HSA formulation can be estimated by comparing the difference in the decline rates of simvastatin and m β -CD between the two cell lines. Compared with Cy5-HSA formulation, Cy5-BPY@HSA formulation presented significant inhibition by simvastatin (79.6% versus 23.3%) but no crucial inhibition on m β -CD (40.5% versus 31.8%) in MCF-7 cell line, demonstrating that the uptake of Cy5-HSA formulation in MCF-7 cells depended more on gp60 whereas Cy5-BPY@HSA formulation not. As a contrast, Cy5-BPY@HSA formulation presented moderate inhibition on simvastatin (77.3% versus 42.1%) and m β -CD (59.9% versus 40.3%), implying that both Cy5-HSA and Cy5-BPY@HSA formulations depended on gp60 in 4T1 cells. **In summary, the uptake of Cy5-HSA formulation depended on gp60 receptor-mediated endocytosis in MCF-7 and 4T1 cell lines, and Cy5-BPY@HSA formulation depended on gp18 and gp30 receptor-mediated endocytosis in MCF-7 cell line, but it depended more on gp18 and gp60 receptor-mediated endocytosis in 4T1 cell line.**

The above discussions have been included in the revised manuscript and Supplementary Note 2. We appreciate the reviewer for his/her valuable comments once again, which greatly helped us understand the exact cellular uptake mechanism. For easier presentation, we summarized the above conclusions using schematic illustration as **Supplementary Fig. 22**.

Supplementary Fig. 22. Schematic illustration of the endocytosis mechanisms of BPY-HSA in **a** 4T1 and **b** MCF-7 cells, and BPY@HSA in **c** 4T1 and **d** MCF-7 cells. BPY@HSA presented superior endocytosis performance than BPY-HSA since it involves more uptake pathways and receptor interactions.

Fig. 3 *In vitro* cellular uptake evaluation and infiltration in 3D tumor spheroids. Fluorescence histograms of **a**, **b** MCF-7 and **c**, **d** 4T1 cells administrated with Cy5-HSA and Cy5-BPY@HSA at

different time, respectively. Cellular uptake mechanism investigations of Cy5-HSA and Cy5-BPY@HSA on **e** MCF-7 and **f** 4T1 cell lines. **g** Receptor interaction investigations of Cy5-HSA and Cy5-BPY@HSA formulations on MCF-7 and 4T1 cell lines. **h** Photoacoustic imaging of cells treated with BPY-HSA or BPY@HSA for 6 h (1×10^6 cells). **i** MCF-7 and **j** 4T1 cellular uptake CLSM images. Z-stack CLSM images of infiltration observation on **k** MCF-7 and **l** 4T1 3D tumor spheroids. Statistical differences p values were represented as * ($p < 0.05$), ** ($p < 0.01$), *** ($p < 0.001$).

Supplementary Fig. 20. Gating strategies for cellular uptake study on **a** MCF-7 and **b** 4T1 cell lines. Representative fluorescence histograms of **c**, **e** MCF-7 and **d**, **f** 4T1 cells administrated with Cy5-HSA and Cy5-BPY@HSA at different conditions, respectively.

Supplementary Fig. 21. Representative fluorescence histograms of the investigation to receptor-mediated endocytosis on **a, b** MCF-7 and **c, d** 4T1 cells administrated with 5 $\mu\text{g}/\text{mL}$ Cy5-HSA and Cy5-BPY@HSA at different conditions, respectively.

5. The authors compare the BPY@HSA nanoparticle system to a BPY-HSA conjugate. Would differences in size between these two systems influence *in vitro* and *in vivo* delivery? This should be mentioned in the work.

Response: We appreciate the reviewer for raising this question on the *in vitro* and *in vivo* delivery of the two systems. The *in vitro* delivery discrepancy of the two systems was discussed in the above response, and we further performed *in vivo* imaging to investigate the fate of the Cy5-HSA and Cy5-BPY@HSA formulations after intravenous administration.

For the MCF-7 xenografted tumor model, the mice were positioned on their side under anesthesia to ensure clear observation of the tumor. As shown in Supplementary Fig. 29, the Cy5-HSA formulation exhibited systemic distribution in regions with high blood flow, which is probably due to that albumin is the major component of the blood. On the other hand, Cy5-BPY@HSA accumulated preferentially in the tumor, liver, and kidney. In particular, the Cy5-BPY@HSA initially entered the liver and reached their maximum concentration at 6 h before gradually being excreted

through the kidneys and gastrointestinal tract, which suggests that the liver and kidneys are the primary organs responsible for the metabolism and excretion of the Cy5-BPY@HSA. Moreover, in the MCF-7 xenograft tumor model, the Cy5-BPY@HSA showed greater accumulation in the tumor with an enhanced trend over a 24-hour period. However, Cy5-HSA demonstrated minimal accumulation in the tumor as the fluorescence signal was indistinguishable from the systemic background. Therefore, we analyzed the extent of accumulation through changes over time and *ex vivo* imaging.

Supplementary Fig. 29. Biodistribution investigation of the Cy5-HSA and Cy5-BPY@HSA formulations on MCF-7 xenografted tumor models. **a** *In vivo* imaging at indicated time points post intravenous injection of Cy5-HSA and Cy5-BPY@HSA (white cycles indicate tumor areas). **b** *ex vivo* imaging of harvested major organs and tumors at 24 h post intravenous injection of Cy5-HSA and Cy5-BPY@HSA. Radiant efficacy of **c** tumor area, **d** selected areas against time, and **e** *ex vivo* organs and tissues.

For 4T1 orthotopic tumor model, the mice were lied under anesthesia to clearly observe the tumor. As shown in Supplementary Fig. 37, Cy5-HSA formulation also presented systemic distribution (the liver is not similar with the side placed mouse in Supplementary Fig. 29, because the height and position difference would influence the acquisition focus of the camera in the instrument) while Cy5-BPY@HSA formulation exhibited preferred distribution in liver, kidney,

gastrointestinal tract and bladder, indicating the liver was the major organ mediating the metabolism of the Cy5-BPY@HSA and GI tract and kidney were the major routes mediating the excretion of the Cy5-BPY@HSA. For the tumor accumulation efficacy, both formulations showed fluorescence signals on tumor sites, and Cy5-BPY@HSA presented enhanced tumor accumulation than that of Cy5-HSA.

Supplementary Fig. 39. Biodistribution investigation of the Cy5-HSA and Cy5-BPY@HSA formulations on 4T1 orthotopic tumor models. **a** *In vivo* imaging at indicated time points post intravenous injection of Cy5-HSA and Cy5-BPY@HSA (white cycles indicate tumor areas). **b** *ex vivo* imaging of harvested major organs and tumors at 24 h post intravenous injection of Cy5-HSA and Cy5-BPY@HSA. Radiant efficacy of **c** tumor area, **d** selected areas against time, and **e** *ex vivo* organs and tissues.

In summary, the tumor accumulation of Cy5-BPY@HSA was found to be superior to that of Cy5-HSA on these two tumor models. Furthermore, moderate tumor accumulation of Cy5-HSA was observed on the 4T1 orthotopic tumor model, as opposed to the MCF-7 xenografted tumor model. It is inferred that the differences in Cy5-HSA formulation between the two models result from the distinct natural characteristics of each tumor model. This result aligns with the findings revealed by the above cellular uptake mechanism experiment results. Additionally, Cy5-BPY@HSA showed a distinct fate compared to natural HSA upon entry into the bloodstream, as it targeted

tumor tissue and was metabolized by the liver before being eliminated through the kidney (via urine) and GI tract (through bile acid or intestinal secretion).

These results and discussions were included in the revised manuscript and supplementary materials.

Modification:

The biodistribution of the two formulations (Cy5-labelled Cy5-HSA and Cy5-BPY@HSA formulations) was observed by fluorescence imaging. As shown in Supplementary Fig. 29, the Cy5-BPY@HSA presented enhanced tumor accumulation whereas Cy5-HSA showed systemic distribution in the whole body, which demonstrated the good tumor target efficacy of Cy5-BPY@HSA (the biodistribution in the body was discussed in Supplementary Note 4).

In addition, the biodistribution of the two formulations (Cy5-labelled Cy5-HSA and Cy5-BPY@HSA formulations) was determined by fluorescence imaging as well. As shown in Supplementary Fig. 39, the Cy5-BPY@HSA presented better tumor accumulation on 4T1 orthotopic tumor model than that of Cy5-HSA, which only showed slight accumulation on MCF-7 xenografted tumor model but exhibited moderate accumulation on 4T1 orthotopic tumor model (the biodistribution in the body was discussed in Supplementary Note 5). This type of profile was consistent with the cellular uptake and 3D tumor spheroid infiltration results. We inferred that such phenomenon might be due to the intrinsic diversity of different tumor models, which is also a common phenomenon in the pharmacology and clinical investigations, and we discussed the possible reasons in the Supplementary Note 6.

Supplementary Note 4: Discussion for Supplementary Fig. 29

For the MCF-7 xenografted tumor model, the mice were positioned on their side under anesthesia to ensure clear observation of the tumor. As shown in Supplementary Fig. 29, the Cy5-HSA formulation exhibited systemic distribution in regions with high blood flow, which is probably due to that albumin is the major component of the blood. On the other hand, Cy5-BPY@HSA accumulated preferentially in the tumor, liver, and kidney. Cy5-BPY@HSA initially entered the liver and reached their maximum concentration at 6 h before gradually being excreted through the kidneys and gastrointestinal tract, which suggests that the liver and kidneys are the primary organs responsible for the metabolism and excretion of the Cy5-BPY@HSA. Moreover, in the MCF-7 xenograft tumor model, the Cy5-BPY@HSA showed greater accumulation in the tumor with an enhanced trend over a 24-hour period. However, Cy5-HSA demonstrated minimal accumulation in the tumor as the fluorescence signal was indistinguishable from the systemic background. Therefore, we analyzed the extent of accumulation through changes over time and ex vivo imaging.

Supplementary Note 5: Discussion for Supplementary Fig. 39

For 4T1 orthotopic tumor model, the mice were lied under anesthesia to clearly observe the tumor. As shown in Supplementary Fig. 37, Cy5-HSA formulation also presented systemic distribution (the

liver is not similar with the side placed mouse in Supplementary Fig. 29, because the height and position difference would influence the acquisition focus of the camera in the instrument) while Cy5-BPY@HSA formulation exhibited preferred distribution in liver, kidney, gastrointestinal tract and bladder, indicating the liver was the major organ mediating the metabolism of the Cy5-BPY@HSA and GI tract and kidney were the major routes mediating the excretion of the Cy5-BPY@HSA. For the tumor accumulation efficacy, both formulations showed fluorescence signals on tumor sites, and Cy5-BPY@HSA presented enhanced tumor accumulation than that of Cy5-HSA.

Considering the results in Supplementary Fig. 29, the tumor accumulation of Cy5-BPY@HSA was found to be superior to that of Cy5-HSA on these two tumor models. Furthermore, moderate tumor accumulation of Cy5-HSA was observed on the 4T1 orthotopic tumor model, as opposed to the MCF-7 xenografted tumor model. It is inferred that the differences in Cy5-HSA formulation between the two models result from the distinct natural characteristics of each tumor model, which would be discussed in Supplementary Note 6. This result aligns with the findings revealed by the above cellular uptake mechanism experiment results. Additionally, Cy5-BPY@HSA showed a distinct fate compared to natural HSA upon entry into the bloodstream, as it targeted tumor tissue and was metabolized by the liver before being eliminated through the kidney (via urine) and GI tract (through bile acid or intestinal secretion).

6. Figure 2, LE and LC abbreviations in the figure caption should be also be written in full. What does the insert in 2b mean to show?

Response: We appreciate the reviewer for bringing these issues to our attention. We have revised the figure caption to provide greater clarity: The inserted image in Figure 2b is of the corresponding sample that was exposed to a red laser from a vertical orientation. The Tyndall effect observed in the solution is indicative of the heterogeneous distribution of nanoparticles within the solution.

Fig. 2 Characterization and photothermal performance of BPY-HSA and BPY@HSA. **a** TEM images of BPY@HSA. **b** DLS size distribution of BPY@HSA aqueous solution (insert: photograph of BPY@HSA aqueous solution and observed Tyndall effect). **c** Zeta potentials of BPY, BPY-HSA, BPY@HSA aqueous solutions. **d** Loading efficiency (LE) and loading capacity (LC) determination of BPY-HSA and BPY@HSA. **e** UV-vis-NIR absorption spectra of BPY (in water containing 1% DMSO), BPY-HSA and BPY@HSA aqueous solutions. **f** Photoacoustic imaging results of BPY, BPY-HSA, BPY@HSA with indicated concentrations. Photothermal temperature elevation curve of **g** BPY-HSA and **j** BPY@HSA under 1 W/cm² laser irradiation (wavelength = 808 nm). Photothermal stability of **h** BPY-HSA and **k** BPY@HSA under 1 W/cm² laser irradiation (wavelength = 808 nm). PCE

determination of i BPY-HSA and I BPY@HSA.

7. In the in vitro delivery work, what cellular receptors do the authors suggest albumin interacts with? Figure 5, the BPY@HSA and BPY-HSA seemed to elicit similar cytotoxicity in the presence of a NIR laser (Figure 3c and d), but the authors state “BPY@HSA+L group exhibited higher apoptosis rates than that of BPY-HSA+L group in both MCF-7 and 4T1 cells, which might be the result of the enhanced cellular uptake of BPY@HSA.” Can the authors comment and include numbers in the text to support the statement.

Response: Thanks for the reviewer’s kind suggestions.

(7-1) According to the cellular uptake behavior investigation in Fig. 3a-d, the Cy5-HSA and Cy5-BPY@HSA exhibited different uptake behaviors in the cells, and the Cy5-BPY@HSA presented higher uptake level in both cell lines, which were the results of differences in uptake mechanisms and receptors interacted with the formulations (Fig. 3e-g, the details were discussed in Response 4). According to the receptor interaction investigation in Fig. 3g (shown in the answers to question #4 above), the uptake of Cy5-HSA formulation depended on gp60 receptor-mediated endocytosis in MCF-7 and 4T1 cell lines, and Cy5-BPY@HSA formulation depended on gp18 and gp30 receptor-mediated endocytosis in MCF-7 cell line, and it depended more on gp18 and gp60 receptor-mediated endocytosis in 4T1 cell line.

(7-2) The apoptosis rates of each treatment were listed in Supplementary Fig. 26 and 28, and the BPY@HSA+L groups (95.8% for MCF-7 and 97.3% for 4T1) exhibited higher apoptosis rates than that of BPY-HSA+L groups (59.9% for MCF-7 and 82.3% for 4T1). The cellular uptake investigation (Fig. 3a-d) revealed that both cell lines presented enhanced uptake to BPY@HSA, and the 4T1 also showed preferred uptake to BPY-HSA than MCF-7. Meanwhile, the level of uptake efficacies of these groups were corresponded with the level of apoptosis rates. Therefore, we can conclude that the intracellular irradiated PTAs could generate higher cytotoxicity than that of extracellular PTAs. As suggested, we have supplemented the discussion and added the numbers of apoptosis ratio in the revised manuscript for better clarity.

	Control	Control + L	BPY -HSA	BPY -HSA + L	BPY@HSA	BPY@HSA + L
Live (%)	94.4	89.2	94.6	25.3	95	2.01
Early apoptosis (%)	3.03	4.18	3.13	19.1	3.24	11.4
Late apoptosis (%)	2.09	5.24	1.49	40.8	1.53	84.4
Necrosis (%)	0.51	1.34	0.74	14.8	0.23	2.13

Supplementary Fig. 26. Percentage analysis of MCF-7 apoptosis with different treatments indicated.

	Control	Control + L	BPY -HSA	BPY -HSA + L	BPY@HSA	BPY@HSA + L
Live (%)	93.9	92.3	94.9	17.5	94.3	0.26
Early apoptosis (%)	1.28	5.46	2.92	19.4	2.92	34.7
Late apoptosis (%)	4.56	2.2	2.03	62.9	2.77	62.6
Necrosis (%)	0.31	0.075	0.15	0.18	0.039	2.43

Supplementary Fig. 28. Percentage analysis of 4T1 apoptosis with different treatments indicated.

Modification:

The BPY@HSA+L groups exhibited higher apoptosis rates (95.8% for MCF-7 and 97.3% for 4T1) than that of BPY-HSA+L group (59.9% for MCF-7 and 82.3% for 4T1) in both cell lines (Supplementary Fig. 26 and 28), which might be the result of the enhanced cellular uptake of BPY@HSA (Fig. 3a-d), because the intracellular PTAs did cause stronger photothermal anti-tumor effects than that of extracellular PTAs (Supplementary Note 3).

Supplementary Note 3: Discussion for Supplementary Fig. 26 and 28

The apoptosis rates of each treatment were given in Supplementary Fig. 26 and 28, and the BPY@HSA+L groups (95.8% for MCF-7 and 97.3% for 4T1) exhibited higher apoptosis rates than that of BPY-HSA+L groups (59.9% for MCF-7 and 82.3% for 4T1). The cellular uptake investigation (Fig. 3a-d) revealed that both cell lines presented enhanced uptake to BPY@HSA, and the 4T1 also showed preferred uptake to BPY-HSA than MCF-7. Meanwhile, the order of uptake efficacies of these groups (Cy5-BPY@HSA on 4T1 > Cy5-BPY@HSA on MCF-7 > Cy5-HSA on 4T1 > Cy5-HSA on MCF-7) were corresponded with the order of apoptosis rates. Therefore, we can conclude that the intracellular irradiated PTAs could generate higher cytotoxicity than that of extracellular PTAs.

8. Albumin is used as a drug half-life technology, it would be relevant to show the half-life for the BPY@HSA nanoparticle system.

Response: We thank the reviewer's for raising the question about the half-life of the BPY@HSA nanoparticles. We have determined the blood concentration of BPY by ICP-MS, and fitted the blood concentration data with the formula $\ln C = -kt + \ln C_0$. As shown in Supplementary Fig. 30, the half-life of BPY@HSA nanoparticles was calculated as 7.64 h.

Modification:

Furthermore, the blood concentration changes against time (Supplementary Fig. 30) were also determined to evaluate the pharmacokinetic property of BPY@HSA, and the half-life of BPY@HSA was calculated as 7.64 h in balb/c nude mice.

Supplementary Fig. 30. Blood concentration of BPY@HSA changes against time ($n = 3$).

9. The number of animals are not mentioned in the figures 5 and 6. This is important to include. In the supplementary 5 animals is cited, is this the number used in figure 5 and 6.

Response: We thank the reviewer's advice. The number of animals participated in the tumor treatments procedures is 5 for each treatment group. We have appended the biological replicates $n = 5$ to the captions of Fig. 5 and 6 in the revised manuscripts as follows.

Modification:

Fig. 5 Photothermal therapy on xenografted tumor models. a Schematic illustration of treatments procedures ($n = 5$). b Photoacoustic imaging results of tumor sites of BPY-HSA and BPY@HSA at different time points ($n = 1$). c Photothermal imaging results of mice administrated with PBS, BPY, BPY-HSA, BPY@HSA after 808 nm laser irradiation (1 W/cm^2). d *In vivo* photothermal temperature elevation curves of different groups ($n = 5$). e Tumor growth curve of each group ($n = 5$). f Body weight of the mice during the treatment. g Digital photo of harvested tumors. h Representative slides of H&E staining, TUNEL and Ki67 staining of the tumor tissues after different treatments. Statistical differences p values were represented as * ($p < 0.05$), ** ($p < 0.01$), * ($p < 0.001$).**

Fig. 6 Photothermal therapy on orthotopic tumor models. **a** Schematic illustration of treatments procedures ($n = 5$). **b** Photoacoustic imaging results of tumor sites of BPY-HSA and BPY@HSA at

different time points ($n = 1$). **c** Photothermal imaging results of mice administrated with PBS, BPY, BPY-HSA, BPY@HSA after 808 nm laser irradiation (1 W/cm^2). **d** *In vivo* photothermal temperature elevation curves of different groups ($n = 5$). **e** Tumor growth curve of each group ($n = 5$). **f** Body weight of the mice during the treatment. **g** Digital photo of harvested tumors. **h** Representative slides of H&E staining, TUNEL and Ki67 staining of the tumor tissues after different treatments. **i** Representative digital photos and H&E staining slides of harvested lungs. **j** Histogram of metastatic nodules in each group ($n = 5$). Statistical differences p values were represented as * ($p < 0.05$), ** ($p < 0.01$), *** ($p < 0.001$).

10. Why did BPY-HSA have a therapeutic effect in the orthotopic model (Figure 6) but not the xenograft model (Figure 5)?

Response: Thanks for pointing out the therapeutic effect difference of BPY-HSA+L group between the orthotopic and xenograft tumor model. The orthotopic tumor model was established by 4T1 cell line whereas the xenograft tumor model was established by MCF-7 cell line. Therefore, the major reason that caused the difference of the therapeutic effect was the inert characterizations of different kinds of tumors. We would like to discuss this concerns in molecular, cellular, and tissue levels.

First of all, it was evidenced that MCF-7 cell line does not possess caveolae-mediated pathway owing to the lack of caveolins and cavins-related molecular machinery (e.g., 10.1038/s41565-021-00858-8; 10.7554/eLife.01434), which is an important pathway for cellular uptake and it also mediates transcellular transport from blood vessels to deep tissues.

Secondly, the cellular uptake to Cy5-HSA (or BPY-HSA) on the levels and mechanisms is different between MCF-7 and 4T1 cell lines. The Cy5-HSA formulation showed different cellular uptake behavior in MCF-7 and 4T1 cell lines (Fig. 3a and 3c), and the 4T1 showed 85.4-fold higher MFI (compare to 0 h group) while MCF-7 showed 59.5-fold (Supplementary Fig. 19), indicating the 4T1 showed preferred uptake to Cy5-HSA formulation than MCF-7. In addition, the CLSM and PA imaging results also indicated that the Cy5-HSA (or BPY-HSA) presented higher cellular uptake level in 4T1 cell lines. In order to clarify the detailed cellular uptake pathways, we performed the cellular uptake mechanism investigation. As shown in Fig. 3e-f, the Cy5-HSA formulation presented different uptake behavior under the pre-treatment of different inhibitors, such as m β -CD, Genistein, and Dynasore. These results indicated that the MCF-7 cell line may more rely on dynamin-dependent FEME pathway, whereas 4T1 cell line may rely more on dynamin-independent caveolae-mediated endocytosis (when comparing with MCF-7). Moreover, the Cy5-BPY@HSA formulation broadened the receptors that could interact with the formulations, which could interact with gp18, gp30, and gp60 at the same time (whereas Cy5-HSA could only interact with gp60). Such receptor interaction variances promoted the cellular uptake to our Cy5-

BPY@HSA formulation as well.

Furthermore, the infiltration ability of Cy5-HSA is different on MCF-7 and 4T1 3D tumor spheroids. Cy5-HSA showed better infiltration ability in 4T1 models than that of MCF-7 (Fig. 3j-k shown above, Supplementary Fig. 19), which might be resulted by the molecular machinery differences between the two cell lines. The lack of caveolae not only caused the low uptake of Cy5-HSA formulation, but also limited the transcellular transport leading the infiltration to deep tissues. Moreover, the tumor accumulation of the Cy5-HSA on 4T1 orthotopic tumor model was much more higher than that of MCF-7 xenograft tumor model (Supplementary Fig. 29 and 39), which might be the result of transcellular transport mediated by caveolae, because the transportation into the deeper tissues greatly helped in the accumulations of the drugs in the tumor sites. Adequate tumor accumulations are the base of good photothermal therapeutic effect. Therefore, the BPY-HSA could exert better photothermal effect on 4T1 orthotopic tumor model.

In a word, the molecular machinery resulted in the difference of the two cell lines on cellular uptake levels and mechanisms, which determined the distinguished effect on the cell, tumor spheroid, and tissue levels. We have discussed these reasons in the revised manuscript and supplementary materials.

Modification:

Supplementary Note 6: Discussion for the accumulation variation of Cy5-HSA (or BPY-HSA) on different tumor models

Cy5-BPY@HSA presented excellent tumor accumulation on two tumor models, and the Cy5-HSA (BPY-HSA) showed moderate tumor accumulation on 4T1 orthotopic tumor model but MCF-7 xenografted tumor model not. The major reason that caused the difference of the therapeutic effect was the inert characterizations of different kinds of tumors, which varied in molecular, cellular, and tissue levels.

On the molecular level, it was evidenced that MCF-7 cell line does not possess caveolae-mediated pathway owing to the lack of caveolins and caveolin-related molecular machinery³, which is an important pathway for cellular uptake and it also mediates transcellular transport from blood vessels to deep tissues. The cellular uptake to Cy5-HSA (or BPY-HSA) on the levels (4T1 could uptake more) and mechanisms (4T1 depended on CLIC and caveolin, while MCF-7 only depended on CLIC) are different between MCF-7 and 4T1 cell lines (discussed in Supplementary Note 2), which caused the uptake difference. Furthermore, the infiltration ability to deep tumor tissues was investigated by 3D tumor spheroids as well. Cy5-HSA showed better infiltration ability in 4T1 models than that of MCF-7 (Fig. 3j-k, Supplementary Fig. 19), which was resulted by the molecular machinery differences and the cellular uptake levels between the two cell lines; that is, the transcytosis ability would influence the infiltration of the formulations, however, only adequate cellular uptake of surface cells on 3D tumor spheroids to the formulation could ensure the deep

infiltration. Not only a series of tumor inert factors affecting the uptake and transcytosis of the formulations in *in vitro* tumor uptake and infiltration, but the results of *in vivo* tumor accumulation also demonstrated the differential tumor target ability of the Cy5-HSA (or BPY-HSA) on the different tumor models. Sufficient tumor accumulation *in vivo* was the base of good photothermal therapeutic effect. Therefore, the BPY-HSA could exert better photothermal effect on 4T1 orthotopic tumor model than that of MCF-7. Even though the BPY-HSA presented moderate tumor accumulation on 4T1 orthotopic tumor model, the photothermal therapeutic effect was still inferior to the BPY@HSA formulations because the BPY@HSA presented much more higher tumor accumulation *in vitro* and *in vivo*.

Supplementary Fig. 29. Biodistribution investigation of the Cy5-HSA and Cy5-BPY@HSA formulations on MCF-7 xenografted tumor models. **a** *In vivo* imaging at indicated time points post intravenous injection of Cy5-HSA and Cy5-BPY@HSA (white cycles indicate tumor areas). **b** *ex vivo* imaging of harvested major organs and tumors at 24 h post intravenous injection of Cy5-HSA and Cy5-BPY@HSA. Radiant efficacy of **c** tumor area, **d** selected areas against time, and **e** *ex vivo* organs and tissues.

Supplementary Fig. 39. Biodistribution investigation of the Cy5-HSA and Cy5-BPY@HSA formulations on 4T1 orthotopic tumor models. **a** *In vivo* imaging at indicated time points post intravenous injection of Cy5-HSA and Cy5-BPY@HSA (white cycles indicate tumor areas). **b** *ex vivo* imaging of harvested major organs and tumors at 24 h post intravenous injection of Cy5-HSA and Cy5-BPY@HSA. Radiant efficacy of **c** tumor area, **d** selected areas against time, and **e** *ex vivo* organs and tissues.

11. Photoacoustic should be described in the introduction. The types of albumin drug delivery approaches e.g. non-covalent association, conjugation, nanoparticles, should be more clearly explained in the introduction.

Response: Thanks for the valuable suggestion. We have introduced photoacoustic properties and described these delivery approaches more clearly in the introduction. Considering the logicity and concise expression, we have briefly discussed the HSA-based drug delivery approaches (e.g. non-covalent association, conjugation, nanoparticles) in the introduction section of the revised manuscript and listed representative examples in Supplementary Table 1.

Modification:

For example, Lu et al. recently reported a SN38-based albumin-binding prodrug, which was prepared through the Michael addition reaction between the maleimide group on SN38 prodrug molecules and the thiol group at the cysteine-34 residue¹². Xie and coworkers developed a

conjugated polymer that mimics fatty acids and can effectively interact with HSA to form nanoparticles via hydrophobic interaction¹³. Other than loading drugs directly, assembling HSA molecules to form HSA-based nanoparticles is also an emerging technique to fabricate HSA-based platforms^{10,11,14,15}. Such approach involves the use of crosslinking agents, such as glutaraldehyde, to prepare drug delivery materials with nanoscale dimensions by chemically crosslinking different HSA molecules together¹⁶.

In addition, the photothermal effect from photothermal therapy not only kills cancer cells but also generates sound waves that can provide additional diagnostic information through the process of photoacoustic imaging²⁵.

Supplementary Table 1 Loading capacity (LC) of several representative HSA-based platforms.

Type	Interaction	Drug	Platform	LC
Covalent binding ^a	-COOH, -NH ₂	Ce6	HSA-Ce6 ²	1.28%
		Oxa(IV)	HSA-Oxa(IV) ²	1.82%
		CySCOOH	HSA-CySCOOH ³	4.1%
	-SH	monomethyl auristatin E	HSA-MMAE ⁴	2%
Noncovalent binding ^b	Hydrophobic interaction	ICG	ICG@HSA-AZO ⁵	3.73%
		PTX	HSA-PTX-RGD ⁶	6.3%
HSA-assembled nanoparticles ^c	Programmed assemble	ICG	HSA-ICG NPs ⁷	11.0%
	Self-assemble	dc-IR825, gambogic acid	HSA/IR825/GA NPs ⁸	1.1%, 0.2%
HSA-based polymers ^d	Modified residues on denatured HSA	DOX	denatured HSA chain-based polymer ⁹	3.91%
HSA-crosslinked nanoparticles ^e	Genipin crosslinked	PTX	Genipin crosslinked HSA fragments ¹⁰	7.0- 8.3%
	Glutaraldehyde crosslinked	Sunitinib analogue	GA crosslinked HSA NP ¹¹	4.0%
	Cysteine crosslinked	PTX	Cysteine crosslinked albumin NP ¹²	18.3%

^a Covalent binding formulated HSA-platforms are based on covalent bonds between drugs and HSA, thus covalent binding strategy requires the drugs to have reactive groups with sulfhydryl groups or amino groups that are naturally existed on HSA protein, and the drugs containing sulfhydryl groups, maleimide groups, and NHS esters could be loaded to HSA by covalent binding.

^b Noncovalent binding formulating strategies are based on hydrophobic interaction, electrostatic attraction, aromatic stack, and some weak bonds, which is not requiring the drugs to own covalent binding groups. Such formulating strategies could be employed to load hydrophobic small molecular drugs, charged molecular drugs, and some biomolecules.

^c HSA-based self-assembled nanoparticles are prepared by noncovalent interactions of HSA and drug units, which own hydrophilic parts and hydrophobic tails that could provide larger hydrophobic space for efficient drug loading.

^d HSA-based polymers are kinds of HSA polypeptide chain derivatives, which are prepared by modifying residues on denatured HSA polypeptide chains. Such polymers enlarged the molecular binding sites for loading more drugs, however, the carrier skeleton would gain the weight as well.

^e HSA-crosslinked nanoparticles require crosslink agents to serve as “glues” to bind the drugs and HSA proteins to form nanoparticles, which relates both covalent and noncovalent interactions among the agents, drugs, and HSA proteins.

12. Figure 1, the BPY-Mal₂ molecule should be clearly defined, does purple depict the BPY, and what are the 4 strands from this when it has only 2 maleimide groups.

Response: We thank the comments for pointing out the expression in our figure that may lead misunderstanding. We redesigned the symbol of BPY-Mal₂ molecule to highlight the bi-maleimide structure of the BPY-Mal₂ molecule in the revised figure.

Modification:

Fig. 1 Fabrication of BPY@HSA and its application in tumor treatment. a Fabrication processes of BPY@HSA by the bridging strategy. **b** Therapeutic functions of BPY@HSA. **c** Schematic illustration of photoacoustic imaging and PTT on tumor model.

Supplementary Fig. 13. Comparison of BPY-HSA and BPY@HSA formulations. For BPY-HSA, each

HSA protein owned only a free thiol group, and thus it could only afford one BPY-Mal₂ molecule. For BPY@HSA, all the disulfide bonds were deconstructed and then reconstructed by BPY-Mal₂ to form BPY-Mal₂-bridged nanoparticles with higher loading capacity.

13. In the Material and Methods, what stain was used in TEM, what medium were the nanoparticles in for zeta potential and size measurements? Cell numbers should be stated for the in vitro work, animal numbers should be included and instrumentation for PA and PT imaging mentioned.

Response: Thanks for pointing out the details of the experiments. The sample for TEM was directly prepared on the copper grids with carbon membrane without any further staining, and the medium for zeta potential and size measurements were ultra-pure water for Fig. 2b-c. The cell and animal numbers were included in the revised manuscript, and the instrumentation for PA and PT imaging were also mentioned in the revised Method.

Modification:

Fig. 5 Photothermal therapy on xenografted tumor models. a Schematic illustration of treatments procedures (n = 5). b Photoacoustic imaging results of tumor sites of BPY-HSA and BPY@HSA at different time points (n = 1). **c** Photothermal imaging results of mice administrated with PBS, BPY, BPY-HSA, BPY@HSA after 808 nm laser irradiation (1 W/cm²). **d** *In vivo* photothermal temperature elevation curves of different groups (n = 5). **e** Tumor growth curve of each group (n = 5). **f** Body weight of the mice during the treatment. **g** Digital photo of harvested tumors. **h** Representative slides of H&E staining, TUNEL and Ki67 staining of the tumor tissues after different treatments. Statistical differences *p* values were represented as * (*p* < 0.05), ** (*p* < 0.01), *** (*p* < 0.001).

Fig. 6 Photothermal therapy on orthotopic tumor models. a Schematic illustration of treatments procedures (n = 5). b Photoacoustic imaging results of tumor sites of BPY-HSA and BPY@HSA at different time points (n = 1). **c** Photothermal imaging results of mice administrated with PBS, BPY, BPY-HSA, BPY@HSA after 808 nm laser irradiation (1 W/cm²). **d** *In vivo* photothermal temperature elevation curves of different groups (n = 5). **e** Tumor growth curve of each group (n = 5). **f** Body weight of the mice during the treatment. **g** Digital photo of harvested tumors. **h** Representative slides of H&E staining, TUNEL and Ki67 staining of the tumor tissues after different treatments. **i** Representative digital photos and H&E staining slides of harvested lungs. **j** Histogram of metastatic nodules in each group (n = 5). Statistical differences *p* values were represented as * (*p* < 0.05), ** (*p* < 0.01), *** (*p* < 0.001).

14. The grammar should be addressed in the manuscript.

Response: Thanks for the comments. We have double checked all the sentences in the manuscript

to make sure the expression is correct and concise. Here are some examples.

Modification:

HSA, the most abundant protein found in blood plasma, has been extensively utilized as a highly effective carrier for drug delivery since the advent of Abraxane, the pioneering HSA-based drug delivery system¹⁻⁵. In spite of the recent emergence of several biocompatible carrier materials, few can match the well-established HSA-based therapeutic platforms that are proven in both bench and bedside. So far, HSA-based platforms have demonstrated proficiency in loading hydrophobic molecules, enhancing biocompatibility, extending circulation times, and devising drug delivery systems that are tailored or responsive to specific targets⁶⁻⁸.

Photothermal therapy (PTT), an emerging method that utilizes photo energy to generate heat and induce damage through photothermal agents (PTAs), has shown encouraging therapeutic effects for treating tumors with controllable precision²²⁻²⁸.

Reviewer #3

In this manuscript, the authors reported a bridging strategy to fabricate a novel HSA-based photothermal agent nano-platform. The as-prepared BPY@HSA presented better PTT activity and photoacoustic response than BPY-HSA, which was obtained by modifying the pristine HSA with BPY-Mal₂ directly. The BPY@HSA could effectively accumulate in tumor sites and serve as a safe PTA to photothermally ablate the tumors. Nevertheless, the article is interesting with a wide range of studies, which deserve publication, but in my opinion, not of outstanding significance for publication in Nature Communications.

Response: Thank you for reviewing our manuscript and providing feedback. We have performed literature investigation and some experiments to support our study, and we sincerely hope that you will now accept the revised version of the paper for publication.

Human serum albumin (HSA) has competing advantages in terms of biosafety and biocompatibility. Compared with most of the existing drug delivery materials, HSA-based tumor therapy platforms are superior for clinical translation, including the success of HSA based anti-tumor drugs such as Abraxane in the market. Due to the huge potential of HSA as a drug delivery material, developing new method to create more efficient HSA-based anti-tumor drugs is of great significance. In addition, it is still a challenge to improve their therapeutic performance and scale up their production because of the lack of strategies to realize high drug loading capacity and facile preparation features.

By capitalizing on these bottlenecks, in this manuscript, we report a new method to fabricate an HSA-based photothermal platform (BPY@HSA). The quantity of drug modification sites was increased by cleaving the disulfide bonds in HSA to thiol groups for improving drug loading capacity. The photothermal agent was designed with a bi-maleimide structure, which could bind with the thiol groups while re-crosslinking the HSA segment to afford a nanoscale structure. The drug loading step and crosslinking step were merged in our strategy so that the preparation process was remarkably simplified. The improvement in drug loading and high drug delivery efficiency of BPY@HSA leads to the optimized tumor treatment effects, as revealed by in vivo anti-tumor experiment results. The methodology developed here could also be used to incorporate more bi-maleimide functionalized therapeutic agents into HSA-based platforms.

1. The absorption spectrum of BPY in Fig. 2e was obtained in water containing 1% DMSO. Does it aggregate in the solution. Are the absorption spectra of BPY (in water containing 1% DMSO), BPY-HSA and BPY@HSA at the same concentration. Why are they different.

Response: Thanks for the comment. For the first question, the BPY-Mal₂ is a small molecular PTA which cannot directly dissolve in water, and thus the BPY formulation was prepared by antisolvent

precipitation method (BPY-Mal₂ solved in DMSO was gradually injected into ultra-pure water under sonication to form the BPY formulation). Therefore, BPY was a nano-suspension formulation containing dispersed BPY-Mal₂ particle. Majority of the nano-suspension formulations could not be constantly stable in the aqueous solution owing to the presence of solid-liquid heterogeneous interfaces. To address the problems, solvents or additives are generally formulated to enhance the stability of the nano suspensions, which is a commonly employed methodology in pharmaceutical formulations. We chose DMSO as the cosolvent because it is a commonly used solvent during the preparation process of nanoparticle formulation. Besides, DMSO is also the exact same solvent for preparing BPY-HSA and BPY@HSA. We found that BPY formulation containing 1% DMSO could keep stable for several tens of hours, and it could resuspend to well-dispersed suspensions by sonication if it aggregated due to being stayed for a long time.

For the second question, the BPY (in water containing 1% DMSO), BPY-HSA and BPY@HSA are the different formulations and the loaded photothermal agent BPY-Mal₂ in each formulation are also in different chemical environments. First, the peak of absorption and extinction coefficient could be influenced by the aggregation status of the molecules. As presented in Fig. 2e, the spectra show that the BPY had three absorption peaks at wavelength of 899, 710, and 625 nm. Noteworthy, the peaks of BPY-HSA and BPY@HSA in the infrared and red region shifted to 885 and 884 nm, 703 and 700 nm, respectively. Such blue shift might result from an H-aggregation of BPY-Mal₂ molecules in these two the formulations. Second, in BPY-HSA and BPY@HSA, the maleimide groups were reacted with the thiol groups in the polypeptide chains, which may result in a tiny variation of the chemical structures of the BPY-Mal₂ and also influence their spectral characters. Considering these factors, it is reasonable to assume that the spectral characteristics of these three substances vary slightly.

2. The photothermal conversion efficiencies of BPY-HSA and BPY@HSA are so high, why not a lower laser power density is used. The permissible laser power density of 808 nm laser is far below 1 W/cm².

Response: We appreciate the reviewer's question. We have conducted a narrative survey on photothermal therapy of 808 nm laser on reputable journals before deciding the treatment protocols (such as *Nature Communications*, *Science Advances*, *Science Immunology*, *Matter*, and *Chem*, see Table R1), and the statistics were shown in Fig. R2. 1 W/cm² is one of the most commonly employed laser power density for 808 nm laser-based photothermal treatments. Therefore, we chose power density of 1 W/cm² to perform the treatments, which made our study to be more comparable with other PTT studies.

For the concerns on permissible laser power density, we guess the reviewer mentioned about

maximum permissible exposure (MPE), which refers to the level of laser irradiation to skin or eye that may be exposed without adverse biological changes (ANSI Z136.1-2014). It is an exposure instruction for the safety of operator rather than treatment procedures. The final benefits of laser-involved operations depend on how we use the laser, and a classic example is myopia surgery requires high power density laser to cut cornea. In photothermal tumor therapy, the laser safety could be completely controlled because the tumor presented higher PTA accumulation and the applied laser could be accurately located to tumor sites. Moreover, the 808 nm laser power density of 1 W/cm² is acceptable even through it is higher than recommended MPE that only defined adverse biological changes rather than acceptable side effects.

According to our *in vitro* safety investigation, 1 W/cm² 808 nm laser irradiation alone did not cause obvious damages to the tested cells, as suggested by the CCK8 experiment results (Fig. 4c-d) and Live/dead staining results (Fig. 4e), and it also cannot induce significant cell apoptosis to the cells (Fig. 4f-g). Besides, the *in vivo* photothermal imaging results demonstrated that irradiated area solely treated with 1 W/cm² 808 nm laser irradiation could only heat up to 43 °C (Fig.5 c-d, Fig. 6 c-d). During the course of therapy, the mice in PBS+L group did not exhibit any noticeable indications of weight loss, indicating that their overall quality of life was not compromised. Our further investigations on *in vivo* safety of laser treated groups also indicated that there was no obvious health damage on blood routine (the values were located in healthy ranges) and no toxicity on major organs (Supplementary Fig. 35-38, 45). Therefore, the 808 nm laser power density of 1 W/cm² is endurable for photothermal therapy.

Fig. R2. Power density statistics of photothermal therapy studies published in reputable journals.

Power density of 808 nm laser	Journal	doi	Year
0.8	Nature Communications	10.1038/ncomms13193	2016
0.6	Nature Communications	10.1038/ncomms14998	2017
1	Nature Communications	10.1038/s41467-018-02915-8	2018
1	Nature Communications	10.1038/s41467-018-03473-9	2018
2	Nature Communications	10.1038/s41467-018-03915-4	2018
1	Nature Communications	10.1038/s41467-018-06529-y	2018
5	Nature Communications	10.1038/s41467-019-10056-9	2019
2.5	Nature Communications	10.1038/s41467-019-11235-4	2019
1	Nature Communications	10.1038/s41467-019-11269-8	2019
0.3	Nature Communications	10.1038/s41467-019-12142-4	2019
1	Nature Communications	10.1038/s41467-019-12771-9	2019
1.5	Nature Communications	10.1038/s41467-020-14963-0	2020
1	Nature Communications	10.1038/s41467-020-16513-0	2020
1.5	Nature Communications	10.1038/s41467-020-19945-w	2020
1	Nature Communications	10.1038/s41467-020-20566-6	2020
1	Nature Communications	10.1038/s41467-020-20860-3	2020
1	Nature Communications	10.1038/s41467-021-21436-5	2021
0.65	Nature Communications	10.1038/s41467-021-22594-2	2021
1	Nature Communications	10.1038/s41467-021-24961-5	2021
1	Nature Communications	10.1038/s41467-021-27640-7	2021
1	Nature Communications	10.1038/s41467-022-28920-6	2022
0.8	Nature Communications	10.1038/s41467-022-29572-2	2022
0.5	Nature Communications	10.1038/s41467-022-30721-w	2022
1.2	Nature Communications	10.1038/s41467-022-32837-5	2022
1.5	Nature Communications	10.1038/s41467-022-30306-7	2022
1.5	Nature Communications	10.1038/s41467-023-38451-3	2023
2	Science Robotics	10.1126/scirobotics.aax0613	2019
1.2	Science Immunology	10.1126/sciimmunol.aau6584	2019
1	Science Advances	10.1126/sciadv.1601031	2016
1	Science Advances	10.1126/sciadv.aaw6264	2019
1	Science Advances	10.1126/sciadv.aay6825	2020
2	Science Advances	10.1126/sciadv.aba2735	2020
0.8	Science Advances	10.1126/sciadv.aba3546	2020
1	Science Advances	10.1126/sciadv.abb0020	2020
1	Science Advances	10.1126/sciadv.abb3116	2020
1	Science Advances	10.1126/sciadv.abc4373	2020
1	Science Advances	10.1126/sciadv.abc8733	2020
0.65	Science Advances	10.1126/sciadv.abd7614	2021
1.5	Science Advances	10.1126/sciadv.abe3588	2021
1	Science Advances	10.1126/sciadv.abn1701	2022
0.5	Science Advances	10.1126/sciadv.abn1805	2022
0.3	Science Advances	10.1126/sciadv.abn3883	2022
1	Science Advances	10.1126/sciadv.abq4659	2022
1	Science Immunology	10.1126/sciimmunol.aan5692	2017
1	Matter	10.1016/j.matt.2020.04.022	2020
1	Matter	10.1016/j.matt.2023.01.001	2023
1	Matter	10.1016/j.matt.2019.03.007	2019
1	Chem	10.1016/j.chempr.2019.08.018	2019

Table R1. Power density statistics summarized from recently published papers relating photothermal therapy.

3. Laser irradiation was applied twice, which is not common for tumor photothermal therapy. For most works, administration and laser irradiation are performed only once, you should explain about the operation.

Response: We thank the review's comments on the treatment procedures of our study. We conducted a narrative survey on photothermal therapy (including 808 nm and other NIR laser, see Fig. R3 and Table R2) laser on reputable journals before the tumor treatments, and we found that half of studies employed twice or more photothermal treatments. For studies employed only once photothermal treatment, these studies were generally accompanied with other types of treatments like chemotherapy, immunotherapy, and chemodynamic therapy. Such combinational therapy might exert better therapeutic effect owing to these studies employed various tumor killing mechanisms.

In our study, our treatment procedures only involved in photothermal therapy which killed tumors by heat. Our purpose was to compare the delivery efficiency and photothermal effect of our crosslinked BPY@HSA system to chemical conjugated BPY-HSA system, especially, we found that the BPY@HSA presented enhanced tumor accumulations than BPY-HSA on two tumor models, which showed applications in improving drug delivery efficiency and optimizing therapeutic effect. In addition, the animal tumor models were established on immunodeficiency balb/c nude mice, which cannot induce adaptive immune response to eliminate residue tumors. Therefore, treatments of twice could also decrease the tumor burden of the mice in effective groups. Moreover, the BPY@HSA is a new drug delivery platform, and the PTA employed in this study was to validate the conceptional design of the platform.

Fig. R3. Number of treatments statistics of photothermal therapy studies published in reputable journals.

Number of irradiation treatments	Journal	doi	Year
2	Nature Communications	10.1038/ncomms5712	2014
1	Nature Communications	10.1038/ncomms13193	2016
1	Nature Communications	10.1038/ncomms14998	2017
2	Nature Communications	10.1038/s41467-018-02915-8	2018
1	Nature Communications	10.1038/s41467-018-03473-9	2018
3	Nature Communications	10.1038/s41467-018-03915-4	2018
2	Nature Communications	10.1038/s41467-018-06529-y	2018
2	Nature Communications	10.1038/s41467-019-10056-9	2019
1	Nature Communications	10.1038/s41467-019-11235-4	2019
3	Nature Communications	10.1038/s41467-019-11269-8	2019
2	Nature Communications	10.1038/s41467-019-12142-4	2019
4	Nature Communications	10.1038/s41467-019-12771-9	2019
1	Nature Communications	10.1038/s41467-020-14963-0	2020
4	Nature Communications	10.1038/s41467-020-16513-0	2020
1	Nature Communications	10.1038/s41467-020-19945-w	2020
1	Nature Communications	10.1038/s41467-020-20566-6	2020
1	Nature Communications	10.1038/s41467-020-20860-3	2020
1	Nature Communications	10.1038/s41467-021-21436-5	2021
2	Nature Communications	10.1038/s41467-021-22594-2	2021
3	Nature Communications	10.1038/s41467-021-24838-7	2021
1	Nature Communications	10.1038/s41467-021-24961-5	2021
6	Nature Communications	10.1038/s41467-021-27640-7	2021
1	Nature Communications	10.1038/s41467-022-28920-6	2022
1	Nature Communications	10.1038/s41467-022-29572-2	2022
1	Nature Communications	10.1038/s41467-022-30721-w	2022
10	Nature Communications	10.1038/s41467-022-30713-w	2022
3	Nature Communications	10.1038/s41467-022-32837-5	2022
3	Nature Communications	10.1038/s41467-019-10847-0	2019
2	Nature Communications	10.1038/s41467-022-30306-7	2022
3	Nature Communications	10.1038/s41467-023-38451-3	2023
1	Science Robotics	10.1126/scirobotics.aax0613	2019
1	Science Immunology	10.1126/sciimmunol.aau6584	2019
1	Science Advances	10.1126/sciadv.1601031	2016
3	Science Advances	10.1126/sciadv.aaw6264	2019
1	Science Advances	10.1126/sciadv.aay6825	2020
2	Science Advances	10.1126/sciadv.aba2735	2020
1	Science Advances	10.1126/sciadv.aba3546	2020
1	Science Advances	10.1126/sciadv.abb0020	2020
3	Science Advances	10.1126/sciadv.abb3116	2020
1	Science Advances	10.1126/sciadv.abc4373	2020
1	Science Advances	10.1126/sciadv.abc8733	2020
1	Science Advances	10.1126/sciadv.abd7614	2021
8	Science Advances	10.1126/sciadv.abe3588	2021
1	Science Advances	10.1126/sciadv.abn1701	2022
2	Science Advances	10.1126/sciadv.abn1805	2022
2	Science Advances	10.1126/sciadv.abn3883	2022
3	Science Advances	10.1126/sciadv.abq4659	2022
5	Science Immunology	10.1126/sciimmunol.aan5692	2017
1	Matter	10.1016/j.matt.2020.04.022	2020
1	Matter	10.1016/j.matt.2023.01.001	2023
4	Matter	10.1016/j.matt.2019.03.007	2019
8	Chem	10.1016/j.chempr.2022.06.006	2022
1	Chem	10.1016/j.chempr.2019.08.018	2019

Table R2. Number of irradiation treatments statistics summarized from recently published papers relating photothermal therapy.

4. Why did Cy5-BPY@HSA present deeper penetration depth than that of Cy5-HSA in both 4T1 and MCF-7 tumor spheroids. The surface of Cy5-BPY@HSA and Cy5-HAS should be similar.

Response: We thank the reviewer's comments on the tumor penetration of 3D tumor spheroids. The cellular uptake mechanism and receptor interactions with drugs were the main influence factor to the penetration depth of the nanoparticles. We have performed cellular uptake competition by various uptake inhibitors (or condition) and receptor competitively binding agents to explore the detailed uptake behavior of the two formulations.

The cellular uptake mechanism investigations were shown in Fig. 3e-f. Both Cy5-HSA and Cy5-BPY@HSA formulations presented energy-dependent cellular uptake, and they also presented clathrin-mediated endocytosis (Chloroquine) and macropinocytosis (Amiloride) on the two cells. The uptake efficiency decreased by m β -CD and simvastatin revealed that both formulations could enter the cells through caveolin (4T1) and clathrin-independent/dynamin-dependent pathway since there was no molecular machinery of caveolin in MCF-7 cells. Furthermore, the uptake efficacy decreased by Genistein and Dynasore demonstrated that there exists transcytosis in the MCF-7 and 4T1, which may influence the infiltration of the formulations. Moreover, compared with Cy5-HSA, Cy5-BPY@HSA formulation relied more on receptor-mediated endocytosis (evidenced by Chlorpromazine).

We further explored the receptor interactions between tumor cells and the two formulations. Previous studies revealed that gp18, gp30, and gp60 are the major receptors mediating the endocytosis of HSA and HSA-based materials. Here, we investigated the major receptors that interact with the Cy5-HSA and Cy5-BPY@HSA by simvastatin (block caveolae and clathrin-independent carrier pathways but not block gp60, a compensation of m β -CD), F-alb (formaldehyde-treated albumin, competitively inhibit gp18), and M-alb (maleylated albumin, competitively inhibit gp30), and F-alb and M-alb were prepared by previous reported methods (10.1021/acsami.1c03065). As shown in Fig. 3g, the uptake of Cy5-HSA formulation depended on gp60 receptor-mediated endocytosis in MCF-7 and 4T1 cell lines, and Cy5-BPY@HSA formulation depended on gp18 and gp30 receptor-mediated endocytosis in MCF-7 cell line, and it depended more on gp18 and gp60 receptor-mediated endocytosis in 4T1 cell line. To visualize the differences of endocytosis mechanism of the formulations on different cell lines, we clearly presented these differences in Supplementary Fig. 22.

Considering the differences on the uptake mechanisms and receptor interactions in the two formulations, the infiltration of the two formulations was caused by receptor-mediated transcytosis, and the levels of transcytosis were similar since there was slight uptake efficiency variation on Genistein and Dynasore between Cy5-HSA and Cy5-BPY@HSA groups. **Therefore, the infiltration of the formulations into the inside tumor was closely associated with the uptake**

ability of the tumor cells (Fig. 3a-d), and the enhanced uptake ability of Cy5-BPY@HSA made it possessed better tumor infiltration profiles.

Supplementary Fig. 22. Schematic illustration of the endocytosis mechanisms of BPY-HSA in **a** 4T1 and **b** MCF-7 cells, and BPY@HSA in **c** 4T1 and **d** MCF-7 cells. BPY@HSA presented superior endocytosis performance than BPY-HSA since it involves more uptake pathways and receptor

interactions.

Fig. 3 *In vitro* cellular uptake evaluation and infiltration in 3D tumor spheroids. Fluorescence histograms of **a, b** MCF-7 and **c, d** 4T1 cells administrated with Cy5-HSA and Cy5-BPY@HSA at different time, respectively. Cellular uptake mechanism investigations of Cy5-HSA and Cy5-BPY@HSA on **e** MCF-7 and **f** 4T1 cell lines. **g** Receptor interaction investigations of Cy5-HSA and Cy5-BPY@HSA formulations on MCF-7 and 4T1 cell lines. **h** Photoacoustic imaging of cells treated with BPY-HSA or BPY@HSA for 6 h (1×10^6 cells). **i** MCF-7 and **j** 4T1 cellular uptake CLSM images. Z-stack CLSM images of infiltration observation on **k** MCF-7 and **l** 4T1 3D tumor spheroids. Statistical differences p values were represented as * ($p < 0.05$), ** ($p < 0.01$), *** ($p < 0.001$).

For the review's comment on the surfaces of Cy5-HSA and Cy5-BPY@HSA, we washed the excess drugs before imaging (also see Method part: When the average size of the 3D tumor spheroids reached 100, 150 and 200 μm , the spheroids were incubated with 5 $\mu\text{g}/\text{mL}$ of Cy5-HSA or Cy5-BPY@HSA for 12 h. After that, the spheroids were washed by PBS for 3 times and observed by CLSM, and the cross-section images at different depths were scanned by Z-stack mode). **Therefore, the fluorescence signals on tumor spheroid surfaces of Cy5-HSA and Cy5-BPY@HSA (Fig. 3k-l) were accorded to the cellular uptake ability of the different cells (Fig. 3i-j), which should be corresponded to the cellular uptake levels (Fig. 3 a-d) rather than being similar.**

5. The manuscript was not carefully prepared. Many letters in the figures are obscured.

Response: We appreciate the reviewer's comment. The low resolution of some figures may be due to a software problem. When the WORD document is converted into the PDF document, some original images are compressed resulting in a decrease in resolution. We have done our best to resolve this technical problem in the revised manuscript to ensure the clarity of the figures for better reading experience.

REVIEWERS' COMMENTS

Reviewer #1 (Remarks to the Author):

I prefer to review this paper anonymously.

Reviewer #2 (Remarks to the Author):

The authors have addressed most of the comments, however, a few issues could be addressed.

Original Comment 1 "The authors suggest that the system can be applied for the delivery of other drug types in addition to a photothermal agent, stating in the discussion "The methodology developed here also may be used to incorporate more bi-maleimide functionalized therapeutic agents into HSA-based platforms." A limitation of this system is the requirement for release of certain drug types that need to dissociate from the albumin that may require stimuli-responsive linkers complicating the design. Could the authors comment?"

The work covers a photothermal agent that does not require release from the particles. The authors accept in the rebuttal letter answer to this comment, that stimuli-responsive linkers may be needed for release of alternative active agents that need to be released from the particles. They state "Many research teams have successfully prepared and reported prodrug systems with controllable activation functions by introducing stimuli-responsive linkers into drug molecules. Thus, this technology is relatively mature."

The design of cross-linking may possibly interfere with susceptibility of linkers to cleavability. Stimuli-responsive linker technology that was not investigated and applied in this work. The authors should, therefore, refine the broad claim in the abstract "This work provides a facile strategy to enhance the loading capacity of HSA-based platforms in order to improve delivery efficacy and therapeutic effect.

Also the discussion, "In summary, the bridging strategy holds promising potential for developing HSA-based platforms with high loading capacity and facile preparation feature, and thus it is favorable for further clinical translation. The methodology developed here also may be used to incorporate more bi-maleimide functionalized therapeutic agents into HSA-based platforms." As this implies it will work for drugs that require triggered activated release, when it was not addressed in the work.

Original comment 2 "The authors state in line 129 "When BPY@HSA was incubated in water or in DMEM complete medium (containing 10% of FBS), the size of BPY@HSA showed negligible changes for up to 7 days (Supplementary Fig. 16), indicating the good stability of BPY@HSA, which is important for further in vivo application." The nanoparticles are almost double in size in DMEM and 10 % FBS compared to in water. Does this indicate instability?"

In the rebuttal letter answer, the authors state" Considering there is no doubt that almost all the nanoparticles would interact with proteins after intravenous injection, we employed DMEM containing 10 % FBS to investigate whether the interactions between the BPY@HSA nanoparticles and proteins in the blood circulation would affect the stability of BPY@HSA. As exhibited in Supplementary Fig. 16, the particle size of BPY@HSA increased about 40 nm in DMEM containing 10 % FBS, indicating the protein absorption of the BPY@HSA nanoparticles. This phenomenon is relating to protein corona that is referring to the structure composed of one or more layers of proteins adsorbed on the surface of nanoparticles after entering serum-containing media or body fluids. Besides, the particle size of BPY@HSA in DMEM containing 10 % FBS kept stable in 7 days revealed that such nanoplatfrom could also keep stable when entering to the blood stream." Stability was investigated in only 10% serum, and showed a size increase from ~40 nm to ~80 nm of the nanoparticles (supplementary figure 16). More discussion on possible aggregation of the particles versus type of protein adsorption should be mentioned in the manuscript text.

Original comment 5 "The authors compare the BPY@HSA nanoparticle system to a BPY-HSA conjugate. Would differences in size between these two systems influence in vitro and in vivo delivery? This should be mentioned in the work."

In the rebuttal answer, the authors describe the differences in vivo biodistribution between BPY@HSA and Cy5-HSA. The comment was specifically referring to a BPY@HSA nanoparticle system to a BPY-HSA conjugate. Whilst the Cy5-HSA label conjugate would be a similar size to BPY-HSA conjugate the authors should state the size of the conjugate and comment if any size differences between the conjugate or particle would influence biodistribution.

Original comment 9 "The number of animals are not mentioned in the figures 5 and 6. This is important to include. In the supplementary 5 animals is cited, is this the number used in figure 5 and 6."

This has been addressed. The authors should, however, include the number of animals used in Supplementary figures 29 and 39.

Reviewer #3 (Remarks to the Author):

This revised manuscript could be accepted now.

Response to Reviewers' Comments

Reviewer #1 (Remarks to the Author):

I prefer to review this paper anonymously.

Our response: We appreciate the reviewer's suggestions for our manuscript.

Reviewer #2 (Remarks to the Author):

The authors have addressed most of the comments, however, a few issues could be addressed.

Our response: We thank the reviewer's useful comments for improving our manuscript. We have addressed these issues as shown below.

Original Comment 1 "The authors suggest that the system can be applied for the delivery of other drug types in addition to a photothermal agent, stating in the discussion "The methodology developed here also may be used to incorporate more bi-maleimide functionalized therapeutic agents into HSA-based platforms." A limitation of this system is the requirement for release of certain drug types that need to dissociate from the albumin that may require stimuli-responsive linkers complicating the design. Could the authors comment?" The work covers a photothermal agent that does not require release from the particles. The authors accept in the rebuttal letter answer to this comment, that stimuli-responsive linkers may be needed for release of alternative active agents that need to be released from the particles. They state "Many research teams have successfully prepared and reported prodrug systems with controllable activation functions by introducing stimuli-responsive linkers into drug molecules. Thus, this technology is relatively mature."

The design of cross-linking may possibly interfere with susceptibility of linkers to cleavability. Stimuli-responsive linker technology that was not investigated and applied in this work. The authors should, therefore, refine the broad claim in the abstract "This work provides a facile strategy to enhance the loading capacity of HSA-based platforms in order to improve delivery efficacy and therapeutic effect.

Also the discussion, "In summary, the bridging strategy holds promising potential for developing HSA-based platforms with high loading capacity and facile preparation feature, and thus it is favorable for further clinical translation. The methodology developed here also may be used to incorporate more bi-maleimide functionalized therapeutic agents into HSA-based platforms." As this implies it will work for drugs that require triggered activated release, when it was not addressed in the work.

Our response: We appreciate your valuable comments. As suggested, we have refined our language to avoid broad claims. Here are modifications.

1. In the abstract, we specified the range into "HSA-based crosslinking platforms" to refine the broad claim as following: This work provides a facile strategy to enhance the loading capacity of HSA-based crosslinking platforms in order to improve delivery efficacy and therapeutic effect.

2. In the discussion, we also specified the range into "HSA-based crosslinking platforms", and we tuned the language that may cause broad claim as following: In summary, the bridging strategy holds promising potential for developing HSA-based crosslinking platforms with high loading capacity and facile preparation feature, and thus it is favorable for further clinical

translation.

Original comment 2 “The authors state in line 129 “When BPY@HSA was incubated in water or in DMEM complete medium (containing 10% of FBS), the size of BPY@HSA showed negligible changes for up to 7 days (Supplementary Fig. 16), indicating the good stability of BPY@HSA, which is important for further in vivo application.” The nanoparticles are almost double in size in DMEM and 10 % FBS compared to in water. Does this indicate instability?”

In the rebuttal letter answer, the authors state” Considering there is no doubt that almost all the nanoparticles would interact with proteins after intravenous injection, we employed DMEM containing 10 % FBS to investigate whether the interactions between the BPY@HSA nanoparticles and proteins in the blood circulation would affect the stability of BPY@HSA. As exhibited in Supplementary Fig. 16, the particle size of BPY@HSA increased about 40 nm in DMEM containing 10 % FBS, indicating the protein adsorption of the BPY@HSA nanoparticles. This phenomenon is relating to protein corona that is referring to the structure composed of one or more layers of proteins adsorbed on the surface of nanoparticles after entering serum-containing media or body fluids. Besides, the particle size of BPY@HSA in DMEM containing 10 % FBS kept stable in 7 days revealed that such nanoplatform could also keep stable when entering to the blood stream.”

Stability was investigated in only 10% serum, and showed a size increase from ~40 nm to ~80 nm of the nanoparticles (supplementary figure 16). More discussion on possible aggregation of the particles versus type of protein adsorption should be mentioned in the manuscript text.

Our response: We appreciate your valuable comments. Most of nanoparticles would adsorb proteins after entering the blood, and such size increasing phenomenon was also reported by others (e.g., *Sci. Adv.* 2022, 8, eadd3599, *Nat. Commun.* 2019, 10, 4520, *Nat. Commun.* 2020, 11, 4535, *Nat. Commun.* 2021, 12, 648). Among various proteins, albumin, immunoglobulin, and apolipoprotein are three types of major proteins causing protein adsorption (*Nat. Commun.* 2019, 10, 4520). For our BPY@HSA, the size presented negligible changes in 7 days, indicating that such interaction between proteins and nanoparticles was stable. We discussed the possible aggregation of proteins in the revised Supplementary Fig. 16 and Supplementary Discussion.

Original comment 5 “The authors compare the BPY@HSA nanoparticle system to a BPY-HSA conjugate. Would differences in size between these two systems influence in vitro and in vivo delivery? This should be mentioned in the work.”

In the rebuttal answer, the authors describe the differences in vivo biodistribution between BPY@HSA and Cy5-HSA. The comment was specifically referring to a BPY@HSA nanoparticle system to a BPY-HSA conjugate. Whilst the Cy5-HSA label conjugate would be a similar size to BPY-HSA conjugate the authors should state the size of the conjugate and comment if any size differences between the conjugate or particle would influence biodistribution.

Our response: Thanks for the insightful comments. We further determined the size of the BPY-HSA conjugate and natural HSA by DLS. The particle size of BPY-HSA conjugate is 6.09 ± 0.41 nm, which is slightly larger than that of natural HSA (4.82 ± 0.43 nm). We stated the size differences of HSA, BPY-HSA conjugate, and BPY@HSA nanoparticles in Supplementary Figure 13. We also discussed this point in the manuscript and Supplementary Note 4 in the revised Supplementary Information.

The biodistribution of nanoparticles can be influenced not only by their size, but also by other factors such as shape, charge, nature of particle, and type/location of tumor (Nat. Biotechnol. 2015, 33, 941–951). Besides, the distribution in normal organ/tissues and tumor tissues is also different.

In normal organ and tissues, the size differences do not play a dominant role in the biodistribution of BPY-HSA, because it preserves an intact structure of HSA molecule. Therefore, the biodistribution of BPY-HSA should be similar to that of HSA. Thus, the distribution of BPY-HSA in normal tissues should be consistent with albumin, that is, distributing in sites with sufficient blood flow and accumulating in liver (the albumin is synthesized and recycled by liver). As albumin is a protein, it could not be easily excreted by kidneys. The biodistribution analysis in Supplementary Fig. 29e and 39e presented similar results: the Cy5-HSA formulation primarily accumulated in liver rather than kidney when compared with Cy5-BPY@HSA.

In tumor tissues, the accumulation of therapeutic nanoparticles may be affected by active targeting and the size of particles. The type and location of tumor could also affect the tumor accumulation of nanoparticles. As evidenced by *in vitro* uptake mechanism study, penetrating study in 3D tumor spheroids, and *in vivo* distribution study, the BPY-HSA conjugate formulation presented higher accumulation in 4T1 models (but still lower than nanoparticle formulation). At this point, the accumulation in tumor could be determined by the active uptake to the nanoparticles (influenced by the uptake mechanisms and the receptor-mediated endocytosis). On the other hand, the size could also contribute to the tumor accumulation of the particles. According to the *in vivo* and *ex vivo* imaging in Supplementary Fig. 29 and 39, the nanoparticle formulation presented higher tumor accumulation than conjugate formulation, indicating that only suitable particle size (30-40 nm) could enhance the tumor accumulation, and the particle with ultra-small size is not suitable for tumor accumulation.

In short, the size could influence the biodistribution, while the characteristics and nature of particles as well as type of tumor model would also influence the biodistribution, which resulted in broad distribution of BPY-HSA and enhanced tumor accumulation of BPY@HSA.

Revised **Supplementary Fig. 13**. Comparison of BPY-HSA and BPY@HSA formulations. For BPY-HSA, each HSA protein owned only a free thiol group, and thus it could only afford one BPY-Mal₂ molecule. For BPY@HSA, all the disulfide bonds were deconstructed and then reconstructed by BPY-Mal₂ to form BPY-Mal₂-bridged nanoparticles with higher loading capacity.

Revised **Supplementary Note 4**

For the MCF-7 xenografted tumor model, the mice were positioned on their side under anesthesia to ensure clear observation of the tumor. As shown in Supplementary Fig. 29, the Cy5-HSA formulation exhibited systemic distribution in regions with high blood flow, which is probably due to that albumin is the major component of the blood. On the other hand, Cy5-BPY@HSA accumulated preferentially in the tumor, liver, and kidney. In particular, the Cy5-BPY@HSA initially entered the liver and reached their maximum concentration at 6 h before gradually being excreted through the kidneys and gastrointestinal (GI) tract, which suggests that the liver and kidneys are the primary organs responsible for the metabolism and excretion of the Cy5-BPY@HSA. Moreover, in the MCF-7 xenograft tumor model, the Cy5-BPY@HSA showed greater accumulation in the tumor with an enhanced trend over a 24-hour period, this was related to the particle size differences (Supplementary Fig. 13) and tumor uptake differences (Supplementary Fig. 22) between the two formulation. However, Cy5-HSA demonstrated minimal accumulation in the tumor as the fluorescence signal was indistinguishable from the systemic background. Therefore, we analyzed the extent of accumulation through changes over time and *ex vivo* imaging.

Original comment 9 "The number of animals are not mentioned in the figures 5 and 6. This is important to include. In the supplementary 5 animals is cited, is this the number used in figure 5 and 6."

This has been addressed. The authors should, however, include the number of animals used in Supplementary figures 29 and 39.

Our response: Thank you for pointing this out. We included the number of animals ($n = 1$) used in the figure captions of Supplementary Fig. 29 and 39.

Reviewer #3 (Remarks to the Author):

This revised manuscript could be accepted now.

Our response: We thank the reviewer for the recommendation of publication.